# DeltaFormer: Unlock the State Space of Transformer

**Mingyu Xu**[*]
Seed Team, ByteDance
xumingyu@bytedance.com

**Tenglong Ao**[*]
Seed Team, ByteDance
aotenglong@bytedance.com

**Jiaao He**
Seed Team, ByteDance
hejiaoao@bytedance.com

**Jianqiao Lu**
Seed Team, ByteDance
lujianqiao@bytedance.com

**Guang Shi**[†]
Seed Team, ByteDance
shiguang@bytedance.com

**Shu Zhong**[†,*]
Seed Team, ByteDance
zhongshu@bytedance.com

## Abstract

In recent years, large language models built around the Transformer architecture have achieved breakthrough progress in many fields. At the same time, certain weaknesses in these models have prompted further reflection, with the most fundamental concerns centered on the Transformer architecture itself. The Transformer offers high parallelism and can fully exploit the computing power of GPUs, which has enabled it to replace models such as LSTM over the past few years. However, high parallelism is not a free advantage, as it imposes fundamental limits on model performance. In particular, the problems that the logarithmic-precision Transformer architecture can solve are strictly bounded within the class $TC^0$. Many important tasks are generally considered outside $TC^0$, including Python code execution, entity tracking, chess, and other state-tracking problems. Meanwhile, recent state-space methods based on the Delta Rule have been able to surpass the $TC^0$ limitations of the Transformer, but these approaches suffer from fixed-size state spaces and perform poorly on many tasks. To address this, we re-examine the Transformer from the perspective of a state space with kernel functions, and propose an improved architecture called *DeltaFormer*. We theoretically and empirically demonstrate that this new architecture can overcome the inherent $TC^0$ expressivity limitations of standard Transformers, while remaining at least as effective in language modeling tasks. We hope our work will inspire the design of more expressive models.

## 1 Introduction

In the field of artificial intelligence, the Transformer model [71] has attracted widespread attention for its outstanding performance and broad application prospects since its inception. As the core architecture of modern AI systems, the Transformer has demonstrated remarkable results in various domains [9, 32, 19, 55, 56]. Despite its significant success, the Transformer has shown poor performance on many tasks that lack chain-of-thought reasoning, which has prompted reflection among researchers.

Another noteworthy observation is that existing large models have demonstrated impressive reasoning abilities following reinforcement learning [30, 29, 64]. However, recent studies indicate that RL does not unlock fundamentally new capabilities, and that their abilities remain constrained by the pre-training phase [79, 65]. Since model architecture lies at the core of pre-training, researchers have increasingly turned their attention to its limitations—particularly the expressivity of Transformer

---

[*]Equal contribution.
[†]Corresponding authors.

39th Conference on Neural Information Processing Systems (NeurIPS 2025).

models. Existing Transformer-based models possess restricted expressive power; it has been shown that they belong to the $TC^0$ complexity class [44]. Solving problems in larger classes requires the use of chain-of-thought reasoning [24, 37]. Moreover, numerous real-world tasks go beyond $TC^0$ and fall within $NC^1$, such as entity tracking, Python code evaluation, and chess state tracking [42]. This limitation may be a fundamental reason why current large-scale models exhibit randomness in entity tracking tasks [13]. This naturally raises the question: is it possible to break through the $TC^0$ expressivity barrier that constrains the Transformer?

In the domain of finite state-space methods, recent work has shown that the delta rule can be used to surpass the expressivity limits of $TC^0$ [52, 66]. However, these methods rely on finite-sized state spaces, which inherently face challenges, such as difficulties in performing retrieval tasks [31]. A natural question follows: can we draw inspiration from these approaches to design a more expressive Transformer?

To address this, we re-examine the Transformer architecture through the lens of kernel functions and the delta rule, proposing a new model called *DeltaFormer*. We theoretically and empirically validate that DeltaFormer possesses expressive power exceeding that of standard Transformers, while maintaining comparable performance in language modeling tasks.

The main contributions of this work are as follows:

- We revisit the delta rule from the perspective of kernel functions and propose a new architecture, *DeltaFormer*, which implicitly assigns a state space to the Transformer. In addition, we design a chunk-wise algorithm that enables DeltaFormer to be efficiently implemented in parallel on GPUs.

- We theoretically demonstrate that DeltaFormer has stronger expressivity than the standard Transformer. We prove that by combining the delta rule with a non-linear kernel, a KV cache of size $O(T \log n)$ can track $T$ exchanges among $n$ objects. Furthermore, if the KV cache is compressed every $O(n)$ steps, only $O(n \log n)$ space is required, which is significantly smaller than that needed when using a linear kernel.

## 2  Related Work

**Circuit Complexity**    Boolean circuits have long been used to study parallel complexity. Among the most important complexity classes in this context are $NC$ and $TC$. The $TC$ class focuses on problems solvable by Boolean circuits in which majority gates are the primary operation, while $TC^0$ specifically denotes problems solvable by constant-depth, polynomial-size threshold circuits. For an input of length $n$, previous work has shown that constant-depth Transformers with finite-precision (size $poly(n)$) embeddings can only solve problems in $TC^0$ without chain-of-thought reasoning [44]. Recently, several studies have demonstrated that DeltaNet [63, 77] and related variants [76, 52, 66] can overcome the $TC^0$ complexity limitation inherent to Transformers. These architectures exhibit higher expressivity and achieve improved performance in tasks such as state tracking [27, 66, 42].

**Model Architecture**    Over the past years, researchers have explored various architectures to enhance model capabilities, ranging from early RNNs [46] and LSTMs [69] to today's dominant Transformer [71]. Transformers have quadratic complexity, and numerous efforts have sought to improve their efficiency [51, 28, 68, 5, 54]. Nevertheless, Transformers still possess many valuable properties that are not easily replaced, particularly their ability to perform information retrieval and adapt to forms of dynamic sparsity [31, 3, 47]. Currently, many popular language models [1, 20, 75] continue to use Transformers as the core architecture, while some employ hybrid designs [58, 38, 35]. Our work aims to improve the Transformer by breaking through its expressivity limitations. We summarize the relationship between commonly used model architectures and their parallelism in Figure 1.

**Understanding Transformer**    Since the rise in popularity of the Transformer [71], extensive research has focused on understanding its underlying mechanisms. Several works analyze its powerful approximation capabilities [80, 18, 34], while others explore the dynamics of model training [41, 7, 70] and interpretability [10, 39, 40, 2]. With the advent of large-scale models, increasing attention has been devoted to studying the contextual learning ability of Transformers [25,

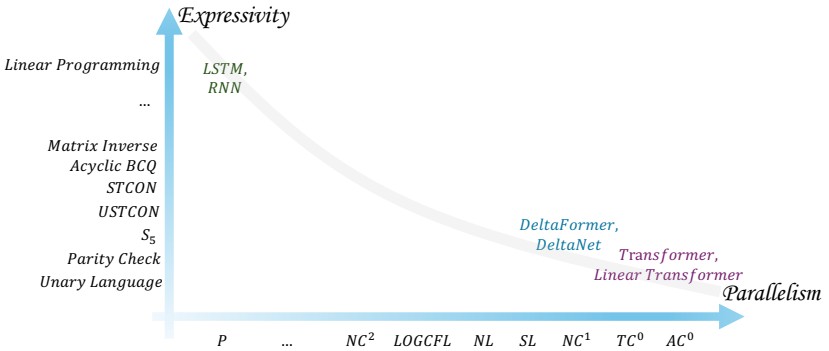

Figure 1: Parallelism and expressivity between models. The higher the parallelism of a model, the more restricted it can be. Previous work has shown that the Log-Precision Transformer is in $TC^0$ [44] and constant precision Transformer in $AC^0$ [37]. The model based on DeltaNet can perform tasks that exceed the expressivity of $TC^0$, if we accept that $TC^0 \neq NC^1$ [52]. In addition, even for models with the same degree of parallelism, their performance is affected by the model capacity. One of the most intuitive examples is that Transformer can perform better than Linear Transformer in retrieval related tasks [31].

36, 49, 23, 21, 22, 73]. An important aspect in understanding the behavior of large models is their associative memory [48, 33, 6, 72]. The delta rule can also be interpreted as an update mechanism for associative memory. In this work, we re-examine the delta rule from the perspective of kernel functions and propose a new model based on this insight.

## 3 Method

### 3.1 Re-examining the Delta Rule from the Perspective of Kernel Functions

The delta rule has a long research history [67, 53, 59] and has recently attracted renewed interest among researchers studying model architectures [63, 77, 52]. Its name comes from updating weights based on the difference between the prediction and the target. Mathematically, let the state space at time $t$ be denoted by $S_t$, the input by $k_t$, and the target by $v_t$. The model seeks to minimize the Euclidean norm $\|S_t k_t - v_t\|_2$. Applying one step of stochastic gradient descent (SGD) with learning rate $\beta_t$ yields:

$$S_{t+1} = S_t - \beta_t(S_t k_t - v_t)k_t^\top \tag{1}$$
$$= S_t \left( I - \beta_t k_t k_t^\top \right) + \beta_t v_t k_t^\top. \tag{2}$$

To retrieve information from $S_t$, we use:

$$o_t = S_t q_t. \tag{3}$$

We now generalize this process by introducing a kernel function $\kappa(x, y) = \psi(x)^\top \psi(y)$, where $x, y \in \mathbb{R}^d$ and $\psi(\cdot)$ is a mapping to an (possibly) infinite-dimensional space. The delta rule with a kernel function becomes:

$$S_{t+1} = S_t \left( I - \beta_t \psi(k_t)\psi(k_t)^\top \right) + \beta_t v_t \psi(k_t)^\top, \tag{4}$$

and information retrieval is performed via:

$$o_t = S_t \psi(q_t). \tag{5}$$

However, this formulation poses two significant challenges. First, both the kernel function $\psi(\cdot)$ and the state matrix $S_t$ are infinite-dimensional, making them infeasible for explicit computation on conventional hardware. Second, updating $S_t$ at each timestep is computationally inefficient; this inefficiency is one of the key reasons why traditional LSTMs have lost popularity in recent years. Therefore, it is essential to develop an efficient GPU-friendly implementation.

## 3.2 Rewriting the Expression for Practical Computation

We can rewrite Eq. 4 in a simplified, equivalent form: $S_{t+1} = S_t + \beta_t u_t k_t^\top$, where $u_t = v_t - S_t \psi(k_t)$ is determined by $\{(k_i, v_i)\}_{i=1}^{t-1}$. Using the derivation in Appendix A, we obtain:

$$u_t = \beta_t v_t - \beta_t \sum_{i=1}^{t-1} \kappa(k_i, k_t)\, u_i, \tag{6}$$

$$o_t = \sum_{i=1}^{t} \kappa(k_i, q_t)\, u_i. \tag{7}$$

Although $\psi(\cdot)$ may map to an infinite-dimensional space, $\psi(\cdot)$ does not explicitly appear in Eq. 6 and Eq. 7, and all terms in these equations can be computed in finite form. If we set $\kappa(x, y) = x^\top y$, this formulation exactly corresponds to DeltaNet [63, 77]. Details on efficient GPU implementation are provided in Section 3.4.

**Generalized Design.** We can generalize the above formulation by applying separate kernel functions $(\kappa_1, \kappa_2)$ in Eq. 6 and Eq. 7, using different gating parameters $(\alpha_t, \beta_t)$ to scale $v_t$ and $\sum_{i=1}^{t-1} \kappa(k_i, k_t)u_i$, and introducing $w_t$ in Eq. 6 to control write and delete operations instead of using $k_t$ directly. This yields:

$$u_t = \alpha_t v_t - \beta_t \sum_{i=1}^{t-1} \kappa_1(k_i, w_t)\, u_i, \tag{8}$$

$$o_t = \sum_{i=1}^{t} \kappa_2(k_i, q_t)\, u_i. \tag{9}$$

If we set $\beta_t = 0$, $\alpha_t = 1$, and define: $o_t = \dfrac{\sum_{i=1}^{t} \exp\left(\frac{k_i^\top q_t}{\sqrt{d}}\right) u_i}{\sum_{i=1}^{t} \exp\left(\frac{k_i^\top q_t}{\sqrt{d}}\right)}$, the formulation degenerates to the standard Transformer. We refer to Eq. 8 and Eq. 9 collectively as *DeltaFormer*. In the following, we compare the advantages of DeltaFormer over previous models such as the Transformer and DeltaNet.

## 3.3 Beyond the Expressivity of Transformer

As previous literature have shown, the performance of Transformer is limited to $TC^0$ if chain of thought is not performed [44, 37, 24]. Based on a hypothesis that is considered correct, $TC^0 \neq NC^1$, We will prove that DeltaFormer can solve a problem which is $NC^1$-complete under $AC^0$ reduction. This task is to track the exchange of $n$ ($n \geq 5$) elements [78]. Specifically, we provide a constructive proof as shown in Theorem 1, and the detailed proof is in Appendix B.

**Assumption 1** *There exist $n$ state points on a $d$-dimensional unit sphere, and the absolute value of the inner product of any two distinct state points is less than or equal to $\epsilon(d, n)$, which means:*

$$\exists\, x_1, x_2, \ldots, x_n \in \mathbb{R}^d \quad s.t. \quad \|x_i\|_2 = 1(\forall i), \quad \max_{i \neq j} |x_i^\top x_j| \leq \epsilon(d, n) < \frac{1}{8}.$$

**Assumption 2** *There is a function $f$ satisfies:*
$$\forall x \in \{-1, 0, 1, 2\}, \forall \tilde{x} \in U(x, 4\epsilon(d, n)): \quad f(\tilde{x}) = x.$$

**Theorem 1** *Based on the above assumptions, we can consider initializing $n$ key-value pairs as $\{(k_1, v_1), \ldots, (k_n, v_n)\}$. The keys $\{k_1, \ldots, k_n\}$ lie on a $d$-dimensional unit sphere and satisfies Assumption 1, which means:*

$$\forall\, i, j \in \{1, \ldots, n\}, i \neq j: \quad \|k_i\|_2 = 1, \quad |k_i^\top k_j| \leq \epsilon(n, d).$$

*and define an attention mechanism as follows:*

$$u_t = v_t - \sum_{i=1}^{t-1} f(k_i^\top k_t) u_i, \quad o_t = \sum_{i=1}^{t} f(q_t^\top k_i) u_i,$$

*where $f(\cdot)$ satisfies Assumption 2 and it is noted that $\forall\, i \in \{1, \ldots, n\}$, since $f(k_i^\top k_i) = 0$, we have $u_i = v_i$.*

*At the current step $t$, $t > n$, the value corresponding to $k_i$ is denoted by $\tilde{v}_i$, $i \in \{1, \ldots, n\}$. Note that, after $t - 1 - n$ exchanges, $\tilde{v}_i$ is not necessarily equal to the initially assigned $v_i$. $\forall\, 1 \le t_2 < t_1 \le n$, to exchange the stored values $\tilde{v}_{t_1}$ and $\tilde{v}_{t_2}$ corresponding to $k_{t_1}$ and $k_{t_2}$, it suffices to construct:*

$$k_t = k_{t_1} - k_{t_2}, \quad v_t = 0$$

*When retrieving the values:*

*Query $q_t = k_{t_1}$, then $o_t = \tilde{v}_{t_2}$;*

*Query $q_t = k_{t_2}$, then $o_t = \tilde{v}_{t_1}$;*

*Query $q_t = k_{t_3}$, $1 \le t_3 \le n$, $t_3 \ne t_1, t_2$, then $o_t = \tilde{v}_{t_3}$.*

*This implies the exchange of values corresponding to $k_{t_1}$ and $k_{t_2}$ is completed.*

### 3.3.1 Re-examining the Assumptions

We now revisit the two assumptions in Theorem 1.

What relationship between $d$ and $n$ must hold for Assumption 1 to be satisfied? According to Theorem 5.2.1 in [82], *For every $\alpha \in (0, 1)$ and $\varepsilon > 0$, there exists $c > 0$ such that for every $d$, one can find at least $2^{cd}$ unit vectors in $\mathbb{R}^d$ whose pairwise inner products all lie in $[\alpha - \varepsilon, \alpha + \varepsilon]$.* Setting $\alpha = 0.01$ and $\varepsilon = 0.1$, we obtain $\epsilon(d, n) \le 0.11 < \frac{1}{8}$. This implies that for Assumption 1 to hold, it is required that: $d = O(\log n)$.

Regarding the choice of $f(\cdot)$ in Assumption 2, one straightforward option is to use the rounding function $\mathrm{round}(\cdot)$, i.e., mapping the input to its nearest integer. With appropriate rounding precision, this function can satisfy the assumption. If we instead adopt the commonly used exponential kernel $\exp(\cdot)$, theoretical results show that a multi-query attention mechanism with four shared heads can express a function $f(\cdot)$ that fulfills Assumption 2.

In summary, according to Theorem 1, delta rule-based no-linear kernel attention can achieve state exchange between historical timesteps $t_1$ and $t_2$, with $d = O(\log n)$.

**Space Cost Analysis** In Theorem 1, suppose we wish to track $T$ exchanges among $n$ states. Clearly, we must set $d = O(\log n)$ and maintain a KU cache of length $T$. This results in a space cost of: $O(T \log n)$.

If we read out $\{\tilde{v}_1, \ldots, \tilde{v}_n\}$ based on $\{k_1, \ldots, k_n\}$ every $O(n)$ steps, and then rewrite $\{(k_i, \tilde{v}_i)\}_{i=1}^n$ into the KU cache, we only require a KU cache of length $O(n)$. This reduces the total space cost to: $O(n \log n)$, which is significantly smaller than the $O(n^2)$ space required when $f(\cdot)$ is an identity mapping. For example, RWKV7 [52] uses a $5 \times 5$ matrix to track the exchange of 5 elements. To provide a clearer comparison between linear and nonlinear kernels, we have restated both versions of Theorem 1 in Appendix C.

For comparison, employing non-linear kernels unlocks the full potential of the delta rule: we can track the exchange of exponentially many states, rather than just the $n = O(d)$ most recent states.

In summary, DeltaFormer—as a generalized form of the Transformer— not only surpasses the inherent $TC^0$ expressivity limitations of the original Transformer, but also tracks the exchange of $n$ objects using significantly smaller state space than models such as RWKV7.

### 3.4 Efficient Chunk-wise Algorithm on GPUs

High parallelism on GPUs has been a key reason why Transformers have outperformed non-parallelizable models such as LSTMs during the scaling era of large models. From a high-level perspective, however, there exists a fundamental tension between parallelism and expressivity. Notably, the parallelizability of DeltaNet is lower than that of a standard Transformer; if reduced to the level of an LSTM, it would lose much of its practical appeal. Therefore, in order to make a more expressive model practically useful, we must devise an implementation that runs efficiently on GPUs.

Consider $q, k, v, u \in \mathbb{R}^d$, a sequence length $T$, and assume $T \gg d$. For simplicity, and without loss of generality, we focus on efficiently computing: $u_t = v_t - \sum_{i=1}^{t-1} \kappa(k_i, k_t) \, u_i$, where the computation of $o_t$ follows similarly to FlashAttention [15].

If we compute $\{u_1, \ldots, u_T\}$ directly according to the above recurrence, the computational complexity is $O(T^2 d)$. Since each $u_t$ depends on all previous $u_{<t}$, this algorithm is inherently sequential—requiring $O(T)$ iterations—and thus cannot exploit GPU parallelism.

**Matrix Formulation.** We can express the recurrence in matrix form as:

$$U = V - AU, \tag{10}$$

where the $t$-th rows of $U$ and $V$ are given by $U_{t,:} = u_t$ and $V_{t,:} = v_t$, respectively. The similarity matrix $A$ is defined by: $A_{t,i} = \kappa(k_t, k_i)$, e.g., $\frac{\exp(k_i^\top k_t)}{Z_t^{(1)}}$ for a normalized exponential kernel, or $k_t^\top k_i$ for a linear kernel. Since $A$ is lower triangular, $U$ can be solved via:

$$U = (I + A)^{-1} V. \tag{11}$$

This formulation requires a matrix inversion, still costing $O(T^2 d)$ operations, but crucially it is fully parallelizable on GPUs.

**Chunk-wise Parallelization.** A compromise between the two approaches above can be achieved via chunk-wise processing. Divide the sequence of length $T$ into $N$ equal-sized chunks, each of length $C = \lfloor T/N \rfloor$. Then, compute $u_t$ within each chunk using parallelizable matrix operations, processing the chunks sequentially.

This reduces the number of recurrent steps from $O(T)$ to $O(N)$, while computational complexity changes from $O(T^2 d)$ to: $O(T^2 d + TCd + TC^2)$. In essence, this method trades additional computation for reduced runtime[3]. For pseudo-code of the chunk-wise implementation, refer to Appendix E.

## 4 Experiment

### 4.1 Track the Exchange of Elements

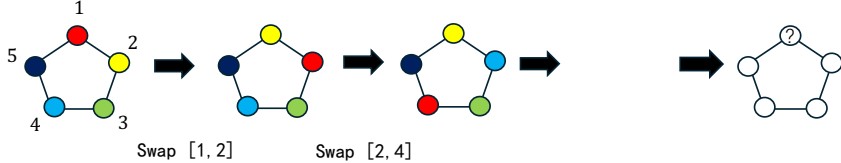

Figure 2: Swap task diagram. At the beginning, tokens of different colors are placed at positions 1 to 5, and the tokens of two positions are exchanged at each step. We expect the model to query what the token for each position is at each step. Simply but without loss of generality, we default to outputting the token at the first position to avoid introducing a "query token". This task can also be tokenized into a task with an input vocabulary size of $C_5^2 = 10$ and an output vocabulary size of $5$.

Although Theorem 1 establishes that DeltaFormer can theoretically track the exchange of $n$ objects, it remains necessary to validate this capability empirically. Specifically, we investigate whether DeltaFormer can learn to track the exchange of $n$ objects from data when trained using gradient descent. To this end, we design an experiment to verify this property. The experimental setup is illustrated in Figure 2, with a default context length of 16.

**DeltaFormer can track the exchange of elements.** We compared DeltaFormer and the standard Transformer under various designs of the similarity function, as shown in Figure 3. Across almost all reasonably simple choices of $\kappa_1(\cdot)$, DeltaFormer achieved better results than the Transformer. In

---

[3]Our code is available at `https://github.com/fla-org/flash-linear-attention/blob/main/fla/layers/deltaformer.py`, and in the supplementary materials.

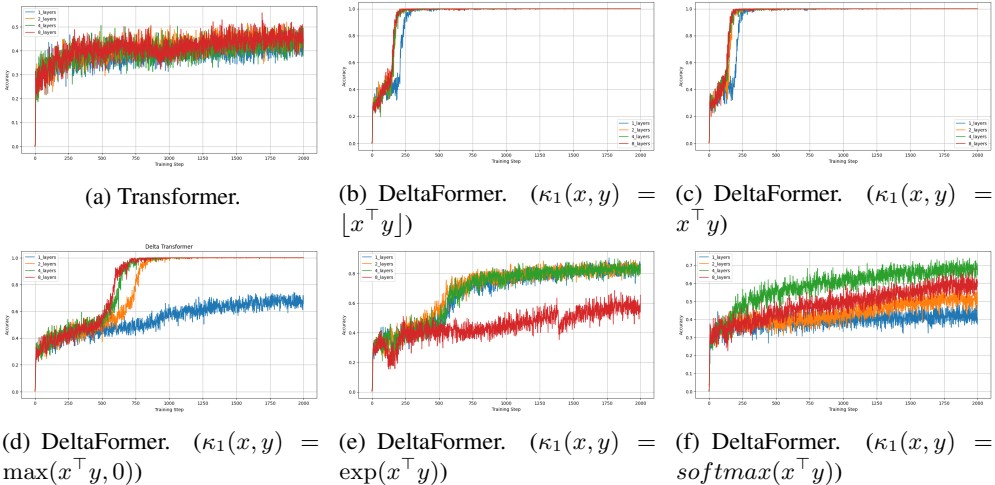

(a) Transformer.

(b) DeltaFormer. $(\kappa_1(x, y) = \lfloor x^\top y \rfloor)$

(c) DeltaFormer. $(\kappa_1(x, y) = x^\top y)$

(d) DeltaFormer. $(\kappa_1(x, y) = \max(x^\top y, 0))$

(e) DeltaFormer. $(\kappa_1(x, y) = \exp(x^\top y))$

(f) DeltaFormer. $(\kappa_1(x, y) = softmax(x^\top y))$

Figure 3: Comparison of Transformer and DeltaFormer using different similarity functions $\kappa_1(\cdot)$ for performing swapping tasks. For $\kappa_2(\cdot)$, we use the softmax function to maintain consistency with Transformer. Pay attention to the scale of the y-axis. To ensure convergence, $\lfloor \cdot \rfloor$ means round to two decimal, such as $\lfloor 1.236 \rfloor = 1.24$.

particular, a 1-layer DeltaFormer was able to execute and track the exchange operations of 5 elements. In contrast, increasing the number of Transformer layers did not yield improvements.

**The similarity function used in Eq. 8 is important for tracking.** Another key observation is that the choice of similarity function has a significant impact on exchange-tracking performance. As shown in our constructive proof in Theorem 1, the appropriately chosen similarity function can track 5 elements with perfect accuracy. The closer the chosen similarity function is to the constructive form, the better the tracking performance. The normalization term in `softmax` negatively impacts the similarity computation when using the exponential function $\exp(\cdot)$. Notably, in our experiments the retrieval similarity function $\kappa_2$ (used in Eq. 9) was not based on the constructive similarity—so as to remain consistent with standard attention—but instead used `softmax`. Even in this case, a suitable choice of $\kappa_1$ can still achieve 100% tracking accuracy. Theorem 1 effectively proves that it is possible to retrieve element values at each position via a specific form of $u$, meaning that the exchange of elements is implicitly captured in the update of $u$.

Intuitively, an inappropriate choice of $\kappa_1$ leads to greater cumulative error in updating $u$. From a mathematical standpoint, this corresponds to the perturbation of an inverse matrix that may be ill-conditioned. For details, see Appendix F. We also performed stress tests using the `Round` and `Linear` kernels in Appendix B.1, tracking exchanges of $n \geq d = 128$ elements.

**The similarity function used in Eq. 9 is important for retrieval.** In the generalized DeltaFormer formulation, the similarity function $\kappa_2$ directly affects retrieval ability. We conducted experiments on the MQAR benchmark [4], with the following configuration: `vocab_size` = 256, `input_seq_len` = 128, `num_kv_pairs` = 32, and `d_model` = 32. To isolate retrieval ability, we used a linear kernel so as not to enhance performance via similarity scaling. The results show that retrieval performance with a purely linear kernel is poor.

| | Linear | Round | ReLU | Softmax |
|---|---|---|---|---|
| **Accuracy (%)** | 85.6 | 91.6 | 99.5 | 99.1 |

Table 1: Impact of $\kappa_2$ on Retrieval Performance

**Curriculum learning is important.** As shown in Figure 4, training directly with a context length of 256 led to very slow convergence.We therefore adopted curriculum learning, starting with length 32 and gradually increasing the window size, i.e., gradually raising task difficulty. Under such a schedule, the model achieved better performance with less computation.

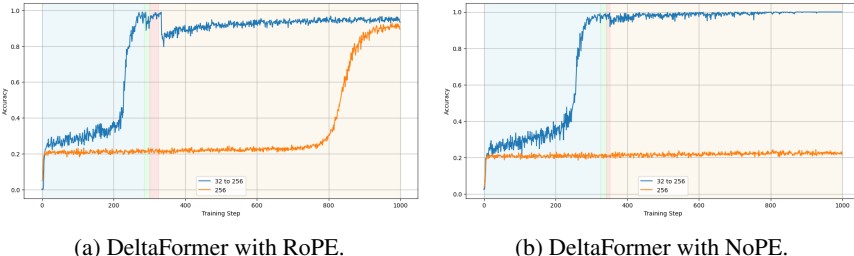

| (a) DeltaFormer with RoPE. | (b) DeltaFormer with NoPE. |

Figure 4: Comparison of DeltaFormer using different learning strategy and position embedding. Each use $\kappa_1(x, y) = \lfloor x^\top y \rfloor$. "32 to 256" means that the initial training length is 32, which means the number of swaps is 32. When the accuracy reaches 0.99, the training length will be doubled until it reaches 256. And "256" means that the model is trained on a training length of 256 from the beginning. The y-axis reflects the accuracy at the current training length.

**The role of rotary embeddings.** Since Theorem 1 does not require positional embeddings, we conducted experiments removing the default rotary position embeddings (RoPE). Without RoPE, convergence slowed and training directly at length 256 yielded random scores. However, under the "32→256" curriculum, accuracy reached $100\%$. Moreover, NoPE models degraded less at jump points during length extension, suggesting better generalization. We speculate that while RoPE may hinder expressivity and extrapolation, it can facilitate optimization.

## 4.2 Reachability of directed acyclic graphs

Furthermore, we design a simple graph connectivity task to evaluate reachability in a directed acyclic graph (DAG). For simplicity, we consider only whether the first node—given a specific topological ordering—can reach other nodes. Initially, each node encodes only its immediate neighbors. Since the final output is binary (`True` or `False`), we avoid class imbalance by dividing the $n$ nodes evenly into two classes and constructing one tree for each class. We then examine reachability from a designated root node, encoding for each node only the information of its parent.

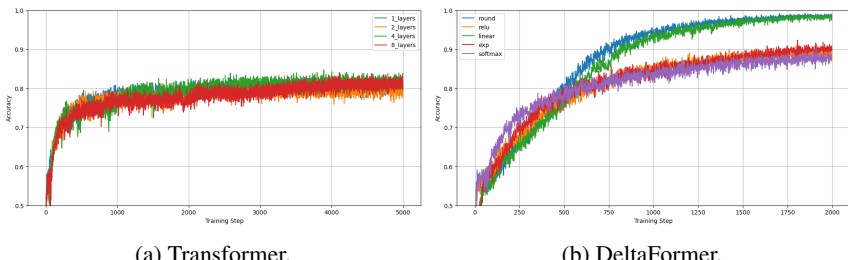

| (a) Transformer. | (b) DeltaFormer. |

Figure 5: Comparison of Transformer and DeltaFormer using different similarity functions $\kappa_1(\cdot)$ for performing swapping tasks. For $\kappa_2(\cdot)$, we use the softmax function to maintain consistency with Transformer. Pay attention to the scale of the y-axis.

**DeltaFormer outperforms Transformer.** We conducted experiments on 32 nodes, as shown in Figure 5. A multi-layer Transformer struggled to reach $100\%$ accuracy, whereas a single-layer DeltaFormer performed highly accurately. In theory, a Transformer requires $O(\log n)$ layers to perform connectivity checks on $n$ nodes [61]. However, based on Figure 5a, we speculate that optimization challenges also limit the Transformer's effectiveness in this task.

**The power of matrix inversion.** As shown in Eq. 11, the computation of $u$ can be rewritten using matrix inversion. If the adjacency matrix $A$ is known, connectivity between nodes $i$ and $j$ can be checked by computing $A, A^2, \ldots, A^n$ and examining whether the $(i, j)$ entry is positive. Since $(I - A)^{-1}$ approximates $I + A + \cdots + A^n$, matrix inversion substantially enhances model expressivity, particularly for graph-related tasks.

**Relation to chain-of-thought (CoT).** The limited depth of Transformers has motivated approaches such as CoT [74, 37] and the Universal Transformer [16, 26], which loop through the layers. In

contrast, our method effectively increases depth along the sequence dimension. This yields higher token efficiency for tasks like DAG connectivity: for a constant-depth Transformer, directed graph reachability requires $O(n^2)$ CoT steps [43], reduced to $O(n)$ with continuous CoT [83], while DeltaFormer achieves the same in a single forward pass.

## 4.3 Language modeling

To verify that DeltaFormer does not affect language modeling capabilities, we conducted experiments on a small scale. Following prior work [76], we use open-source code of them and open-source dataset Fineweb-edu for training and the open-source evaluation tool lm-evaluation-harness for benchmark evaluation. The benchmarks that include LAMBADA [LMB.;[50]], PiQA[8], HellaSwag [Hella.;[81]], WinoGrande [Wino.;[60]], ARC-easy (ARC-e) and ARC-challenge (Arc-c)[12], Boolq [11], OpenbookQA [OBQA.;[45]], SIQA [62] and Copa [57]. We train on a 340M parameter scale with 15B tokens with a peak learning rate of 2e-3. The context length is 2,048 and the global batch size is 0.5M tokens. The experimental results are shown in Table 2.

| Model (340M) | ARC-c acc_n ↑ | ARC-e acc ↑ | Boolq acc ↑ | Copa acc ↑ | Hella. acc_n ↑ | LMB. acc ↑ | OBQA. acc_n ↑ | PIQA acc ↑ | SCIQ. acc ↑ | Wino. acc ↑ |
|---|---|---|---|---|---|---|---|---|---|---|
| Transformer | 28.58 | 59.61 | 60.00 | 68.00 | 40.11 | 34.50 | 38.40 | 67.25 | 81.60 | 52.01 |
| DeltaFormer | | | | | | | | | | |
| $\kappa_1(x,y) = x^\top y$ | 29.01 | 59.09 | 60.52 | 69.00 | 40.43 | 34.17 | 38.20 | 67.90 | 80.60 | 50.04 |
| $\kappa_1(x,y) = Relu(x^\top y)$ | 28.41 | 57.62 | 59.88 | 68.00 | 40.07 | 32.76 | 37.00 | 65.83 | 80.10 | 51.22 |
| $\kappa_1(x,y) = \lfloor x^\top y \rfloor$ | 28.50 | 58.33 | 60.09 | 70.00 | 40.29 | 33.03 | 35.40 | 67.03 | 81.50 | 51.92 |
| $\kappa_1(x,y) = softmax(x^\top y)$ | 28.92 | 57.89 | 61.80 | 69.00 | 40.21 | 34.05 | 37.40 | 67.21 | 81.40 | 52.41 |

Table 2: Comparison of DeltaFormer and its variants on various language modeling benchmarks on the model with 340M parameter.

Due to the fact that at this scale, the fluctuations of one or two points in these benchmark indicators are considered random. Therefore, we can say DeltaFormer is not weaker than standard Transformers in language modeling tasks. Even with different similarity functions, the differences are very small, which is also different from the findings of Section 4.1.

Later, we conducted experiments on the 14B activated MOE model, where the number of key and value was 1/4 of the number of query. As a result, the flops in the self-attention section increased by 25%, and in the entire MOE model, the flops increased by 3%. The results are listed in Table 3 and Table 4.

| | COPA | ARC-E | ARC-C | PIQA | C-Eval | MMLU | RACE-High | RACE-Middle | SIQA | Winogrande | Average |
|---|---|---|---|---|---|---|---|---|---|---|---|
| Transformer | 73.8 | 80.9 | 50.8 | 78.2 | 44.0 | 43.8 | 48.8 | 62.5 | 55.6 | 64.6 | 60.30 |
| DeltaFormer | 74.6 | 82.9 | 51.2 | 79.4 | 46.3 | 45.2 | 49.0 | 62.9 | 54.7 | 67.2 | 61.34 |
| Δ | +0.8 | +2.0 | +0.4 | +1.2 | +2.3 | +1.4 | +0.2 | +0.4 | -0.9 | +2.6 | +1.04 |

Table 3: Benchmark Comparison. We compared the transformer model and delta model with 14B total parameters, and they trained 500 tokens each.

| | General Domain | | Code Domain | |
|---|---|---|---|---|
| Training tokens | Transformer | DeltaFormer | Transformer | DeltaFormer |
| 100 B | 2.06256 | 2.06254 | 1.52628 | 1.48656 |
| 200 B | 1.94337 | 1.94336 | 1.42112 | 1.37757 |
| 300 B | 1.88013 | 1.87554 | 1.35612 | 1.32587 |
| 400 B | 1.83119 | 1.83558 | 1.32285 | 1.29567 |
| 500 B | 1.81472 | 1.81032 | 1.31231 | 1.28326 |

Table 4: Benchmark Comparison. We compared the transformer model and delta model with 14B total parameters, and they trained 500 tokens each.

Firstly, the benchmark results show that the DeltaFormer outperforms the baseline. Then there is the result of training loss, which leads by 0.003 on the general domain, basically aligning with the slight increase of 3% in flops. However, on the code domain, the loss leads by 0.05. When training 300b. The training token can match the baseline training of 400b, which far exceeds the gain of flops. We believe this is due to the higher expressiveness of DeltaFormer, as the code data includes Dyck grammars that match left and right parentheses.

# 5   Discussion

**Expressivity.**   Since matrix inversion lies within the $NC^2$ complexity class [14], the theoretical upper bound of DeltaFormer is likewise in $NC^2$. We can design models with higher expressive power at the cost of reduced parallelism, in the extreme approaching inherently sequential models such as LSTM. Ultimately, a trade-off between parallelism and expressivity must be struck, influenced by hardware and environmental constraints. On this trade-off curve, scaling a model with slightly lower parallelism but higher expressivity than the Transformer may serve as a starting point. Identifying models that fully exploit $NC^2$ may be a promising direction, as many practical models fall within this class, which still allows for parallel execution.

**Optimization.**   Achieving a strong model also depends on optimization. We observed phenomena such as curriculum learning—gradually extending context length—benefiting DeltaFormer in element-tracking tasks. Differences between DeltaFormer and standard Transformers likely lead to distinct optimization behaviors. Furthermore, as discussed in Section 4.1, rotary position embeddings (RoPE) appear to hinder performance but aid optimization. A deeper study of the optimization dynamics of such models is an interesting avenue for future work.

**Scaling.**   Scaling DeltaFormer to larger models and examining the effects will be valuable. We speculate that Transformers require deeper architectures to handle tasks beyond their expressivity, whereas DeltaFormer may achieve such tasks with fewer layers, leading to different optimal depth-to-width ratios. To optimally configure DeltaFormer, original Transformer components may need rethinking. Exploring parameter scaling analogous to Transformer scaling laws is also of interest.

# 6   Limitations

First, although we propose an algorithm that executes efficiently on GPUs, current performance is not optimal and further improvements are needed. Second, our evaluation focuses on toy tasks and small-scale language modeling, without large-scale industrial training to confirm gains for complex tasks. Additionally, a naive layer mixing of linear DeltaNet and standard Transformer could yield a hybrid model with both state-tracking and long-text retrieval abilities. Such a hybrid may have reduced state-tracking capacity but might still suffice for real-world applications. Comparisons between DeltaFormer and simple hybrids thus require validation on practical tasks.

# 7   Conclusion

We extended the delta rule with kernel functions and introduced DeltaFormer. We proved, both theoretically and empirically, that DeltaFormer surpasses the $TC^0$ expressivity limit of Transformers. In particular, introducing nonlinear kernels enables DeltaFormer to track exponentially many element exchanges within the same dimension compared to linear kernels. Experiments indicate that DeltaFormer matches the standard Transformer in language modeling performance. In future work, we aim to scale DeltaFormer to industrial-level training, and hope our findings inspire new Transformer designs with improved expressivity.

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

## A  Delta Rule with Kernel Function

We consider the kernel function $\kappa(x, y) = \psi(x)^\top \psi(y)$, where $\psi(x)$ is a mapping from $d$ to infinite dimensions. Then the delta-rule-based update form and the corresponding read-out equation can be re-write as:

$$S_t = S_{t-1}\big(I - \psi(k_t)\psi(k_t)^\top\big) + v_t\psi(k_t)^\top, \tag{12}$$

$$o_t = S_t\psi(q_t). \tag{13}$$

Hypothesis:

$$S_t = \sum_{i=1}^{t} u_i w_i^\top, \tag{14}$$

where $u_i$ and $w_i$ is pending. Then we have:

$$\sum_{i=1}^{t} u_i w_i^\top = \sum_{i=1}^{t-1} u_i w_i^\top \big(I - \psi(k_t)\psi(k_t)^\top\big) + v_t\psi(k_t)^\top, \tag{15}$$

$$u_t w_t^\top = \sum_{i=1}^{t-1} u_i w_i^\top \big(-\psi(k_t)\psi(k_t)^\top\big) + v_t\psi(k_t)^\top, \tag{16}$$

take the pending $w_i = \psi(k_i)$:

$$
\begin{aligned}
u_t \psi(k_t)^\top &= \sum_{i=1}^{t-1} u_i \psi(k_i)^\top \big(-\psi(k_t)\psi(k_t)^\top\big) + v_t\psi(k_t)^\top \\
&= -\sum_{i=1}^{t-1} \psi(k_i)^\top \psi(k_t) u_i \psi(k_t)^\top + v_t\psi(k_t)^\top \\
&= \left(-\sum_{i=1}^{t-1} \psi(k_i)^\top \psi(k_t) u_i + v_t\right) \psi(k_t)^\top.
\end{aligned}
\tag{17}
$$

Thus, we get the pending $u_t$:

$$
\begin{aligned}
u_t \overline{\psi(k_t)}^\top &= \left(-\sum_{i=1}^{t-1} \psi(k_i)^\top \psi(k_t) u_i + v_t\right) \overline{\psi(k_t)}^\top \\
&= -\sum_{i=1}^{t-1} \kappa(k_i, k_t) u_i + v_t.
\end{aligned}
\tag{18}
$$

Then we have the final update form and the corresponding read-out equation:

$$S_t = \sum_{i=1}^{t} u_i \psi(k_i)^\top = S_{t-1} + u_t \phi(k_t)^\top, \tag{19}$$

$$o_t = \sum_{i=1}^{t} \kappa(k_i, q_t) u_i. \tag{20}$$

## B  Proof of Theorem 1

Before proving Theorem 1, we introduce an auxiliary lemma for facilitating the proof.

**Lemma 1** *Consider Theorem 1, the set of keys $\{k_i\}_{i=1}^{n}$ satisfies Assumption 1, and $k_{>n}$ is the difference between two keys chosen from $\{k_i\}_{i=1}^{n}$. If the function $f(\cdot)$ satisfies Assumption 2, then the following identity holds:*

$$\forall 1 \leq j < i \leq n, \forall l \geq 1: \quad f((k_i - k_j)^\top k_l) = f(k_i^\top k_l) - f(k_j^\top k_l).$$

### B.0.1 Proof of Lemma 1

We distinguish two separate cases according to the value of the index $l$:

**Case 1:** $1 \leq l \leq n$. Consider the following subcases:

    i. If $k_i = k_l$, then we obtain

$$f((k_i - k_j)^\top k_l) = f(1 - k_j^\top k_l) = f(U(1, \epsilon)) = 1$$
$$f(k_i^\top k_l) - f(k_j^\top k_l) = 1 - f(U(0, \epsilon)) = 1.$$

    ii. If $k_j = k_l$, then we have

$$f((k_i - k_j)^\top k_l) = f(k_i^\top k_l - 1) = f(U(-1, \epsilon)) = -1$$
$$f(k_i^\top k_l) - f(k_j^\top k_l) = f(U(0, \epsilon)) - 1 = -1.$$

    iii. If $k_i \neq k_l$, $k_j \neq k_l$, then

$$f((k_i - k_j)^\top k_l) = f(k_i^\top k_l - k_j^\top k_l) = f(U(0, 2\epsilon)) = 0$$
$$f(k_i^\top k_l) - f(k_j^\top k_l) = f(U(0, \epsilon)) - f(U(0, \epsilon)) = 0.$$

**Case 2:** $l > n$. In this case, denote $k_l = k_{l_1} - k_{l_2}$, where $1 \leq l_2 < l_1 \leq n$. Consider the following possibilities regarding the number of equalities among indices $i, j$ and $l_1, l_2$:

    i. If no pair among $(i, j)$ and $(l_1, l_2)$ is equal, then we have

$$f((k_i - k_j)^\top k_l) = f(U(0, 4\epsilon)) = 0$$
$$f(k_i^\top k_l) - f(k_j^\top k_l) = f(U(0, 2\epsilon)) - f(U(0, 2\epsilon)) = 0.$$

    ii. If exactly one pair is equal, we analyze further:

        1. If $i = l_1$, then we have

$$f((k_i - k_j)^\top k_l) = f(U(1, 3\epsilon)) = 1$$
$$f(k_i^\top k_l) - f(k_j^\top k_l) = f(U(1, \epsilon)) - f(U(0, 2\epsilon)) = 1.$$

        2. If $i = l_2$, then we have

$$f((k_i - k_j)^\top k_l) = f(U(-1, 3\epsilon)) = -1$$
$$f(k_i^\top k_l) - f(k_j^\top k_l) = f(U(-1, \epsilon)) - f(U(0, 2\epsilon)) = -1.$$

        3. If $j = l_1$, then similarly

$$f((k_i - k_j)^\top k_l) = f(U(-1, 3\epsilon)) = -1$$
$$f(k_i^\top k_l) - f(k_j^\top k_l) = f(U(0, 2\epsilon)) - f(U(1, \epsilon)) = -1.$$

        4. If $j = l_2$, then similarly

$$f((k_i - k_j)^\top k_l) = f(U(1, 3\epsilon)) = 1$$
$$f(k_i^\top k_l) - f(k_j^\top k_l) = f(U(0, 2\epsilon)) - f(U(-1, \epsilon)) = 1.$$

    iii. If two pairs are equal simultaneously:

        1. If $i = l_1, j = l_2$, we have

$$f((k_i - k_j)^\top k_l) = f(U(2, 2\epsilon)) = 2$$
$$f(k_i^\top k_l) - f(k_j^\top k_l) = f(U(1, \epsilon)) - f(U(-1, \epsilon)) = 2.$$

        2. If $i = l_2, j = l_1$, this contradicts the ordering condition $j < i, l_2 < l_1$ and thus cannot occur.

Combining all the above cases, we have completed the proof.

### B.0.2 Formally Prove Theorem 1

We use mathematical induction to prove Theorem 1.

When $t = n + 1$:

$$k_t = k_{t_1} - k_{t_2}, \tag{21}$$

$$u_t = -\sum_{i=1}^{t-1} f(k_i^\top k_t) u_i = -u_{t_1} + u_{t_2}. \tag{22}$$

If we read the state at $t_1$, i.e., $q_t = k_{t_1}$,

$$\sum_{i=1}^{t} f(q_t^\top k_i) u_i = \sum_{i=1}^{t} f(k_{t_1}^\top k_i) u_i = u_{t_1} + (-u_{t_1} + u_{t_2}) = u_{t_2}. \tag{23}$$

If we read the state at $t_2$, i.e., $q_t = k_{t_2}$,

$$\sum_{i=1}^{t} f(q_t^\top k_i) u_i = \sum_{i=1}^{t} f(k_{t_2}^\top k_i) u_i = u_{t_2} + (u_{t_1} - u_{t_2}) = u_{t_1}. \tag{24}$$

If we read other states, i.e., the state at $j$, where $j \neq t_1, t_2$,

$$\sum_{i=1}^{t} f(q_t^\top k_i) u_i = \sum_{i=1}^{t} f(k_j^\top k_i) u_i = u_j. \tag{25}$$

In summary, at step $t = n + 1$, according to our rules, it is possible to trace the states exchanged between $t_1$ and $t_2$.

Assuming the proposition holds for $t - 1$, we consider the case for $t$ ($t > n + 1$).

At the $t$-th step,

$$k_t = k_{t_1} - k_{t_2}. \tag{26}$$

According to Lemma 1, we have

$$
\begin{aligned}
u_t &= -\sum_{i=1}^{t-1} f(k_t^\top k_i) u_i \\
&= -\sum_{i=1}^{t-1} f(k_{t_1}^\top k_i) u_i + \sum_{i=1}^{t-1} f(k_{t_2}^\top k_i) u_i \\
&= -\tilde{v}_{t_1} + \tilde{v}_{t_2}.
\end{aligned}
\tag{27}
$$

If we read the state at $t_1$, i.e., $q_t = k_{t_1}$,

$$
\begin{aligned}
\sum_{i=1}^{t} f(q_t^\top k_i) u_i &= \sum_{i=1}^{t} f(k_{t_1}^\top k_i) u_i \\
&= \sum_{i=1}^{t-1} f(k_{t_1}^\top k_i) u_i + f(k_{t_1}^\top k_t) u_t \\
&= \tilde{v}_{t_1} + (-\tilde{v}_{t_1} + \tilde{v}_{t_2}) \\
&= \tilde{v}_{t_2}.
\end{aligned}
\tag{28}
$$

If we read the state at $t_2$, i.e., $q_t = k_{t_2}$,

$$\sum_{i=1}^{t} f(q_t^\top k_i)u_i = \sum_{i=1}^{t} f(k_{t_2}^\top k_i)u_i$$
$$= \sum_{i=1}^{t-1} f(k_{t_2}^\top k_i)u_i + f(k_{t_2}^\top k_t)u_t$$
$$= \tilde{v}_{t_2} + (\tilde{v}_{t_1} - \tilde{v}_{t_2})$$
$$= \tilde{v}_{t_1}. \qquad (29)$$

If we read other states, i.e., the state at $j$, where $j \neq t_1, t_2$,

$$\sum_{i=1}^{t} f(q_t^\top k_i)u_i = \sum_{i=1}^{t} f(k_j^\top k_i)u_i$$
$$= \sum_{i=1}^{t-1} f(k_j^\top k_i)u_i + f(k_j^\top k_t)u_t$$
$$= \tilde{v}_j. \qquad (30)$$

In summary, at step $t$, according to our rules, the retrieved states corresponding to $\{k_1, \ldots, k_n\}$ is correct.

By mathematical induction, regardless of how large the exchange step $t$ is, the model can always trace the exchange of $n$ states.

### B.1 The expression of nonlinear vs linear

We also conducted stress tests on the round and linear function in this Section, with $d = 128$. The setting is similar to Section 4.1, but with experiments where $n$ is greater than or equal to 128. We use $n \in \{128, 256, 512\}$, and the corresponding training length is $\{256, 512, 1024\}$ to ensure as much as possible that most elements participate in the exchange. In addition, to avoid optimization issues, we adopted the almost orthogonal vectors used in our Theorem 1 to set key and value of the model and the model only needs to learn to read information from the state space. The results is shown in Figure 6.

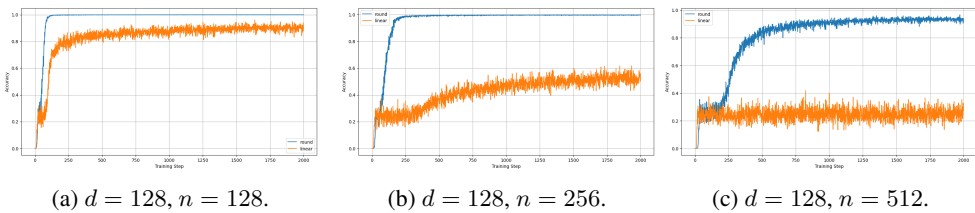

(a) $d = 128, n = 128$.      (b) $d = 128, n = 256$.      (c) $d = 128, n = 512$.

Figure 6: Comparison of DeltaFormer with $\kappa(x, y) = x^\top y$ and $\kappa(x, y) = \lfloor x^\top y \rfloor$.

We can observe that when $d$ is fixed, as $n \geq d$ increases, the performance of the linear kernel is severely degraded. This essentially involves the famous Thompson problem, which is how to place as many orthogonal vectors as possible on the d-dimensional unit sphere. However linear functions cannot have superposition, and nonlinear functions can store a large amount of information through superposition [22].

## C Compare of linear version and no-linear version of Theorem 1

**Theorem 1 (rephrased)** Let the list be $L = [1, 2, \ldots, n]$. Define the swap operation $G_{i,j}$ as

$$G_{i,j}(L)[k] = \begin{cases} L[j], & k = i, \\ L[i], & k = j, \\ L[k], & \text{otherwise}. \end{cases}$$

Let a sequence of swaps be $g_1, g_2, \ldots, g_l \in G = \{G_{i,j} \mid 0 < i < j \le n\}$. Then there exists a single-layer DeltaFormer $F$ with head dimension $O(\log n)$ such that

$$F(g_1, \ldots, g_l, j) = g_1 \circ g_2 \circ \cdots \circ g_l(L)[j].$$

**KU Cache read out and rewrite**

We can introduce the KU cache compress operator $\mathrm{ku}(\cdot)$ that pre-fills the sequence $g_1, \ldots, g_l$ into a cache of size $O(n \log n)$, independent of $l$. With this cache we obtain another single-layer DeltaFormer $H$ satisfying

$$H(\mathrm{ku}(g_1, \ldots, g_l), j) = F(g_1, \ldots, g_l, j).$$

**Comparison with Lemma 2 in RWKV-7**

**Lemma 2 in RWKV-7 (rephrased)** There exists a single-layer RWKV-7 block $F$

- head dimension $O(n)$,
- state-space size $O(n^2)$,

such that

$$F(g_1, \ldots, g_l, j) = g_1 \circ g_2 \circ \cdots \circ g_l(L)[j].$$

# D    What language can DeltaFormer express?

Regarding the expressive power of models, in addition to computational complexity theory, it can also be understood from the perspective of language hierarchy. We conducted an experiment on language hierarchy, and the results are shown in Table 5. The results show that in the category of regular (R) language, Deltaformer has better length generalization ability compared to Transformer, indicating that Deltaformer can learn circuits that are easily length generalized compared to Transformer. But for some more complex languages, such as deterministic context-free (DCF) and context-sensitive (CS) languages, the performance of Deltaformer and Transformer is not as good as RNN models, such as Tape-RNN [17]. However, considering that RNN models have not yet been efficiently implemented on GPUs, we believe that DeltaFormer is practical.

| Level | Task | Model Architecture | | | | | |
|-------|------|------|------|------|------|------|------|
| | | **RNN** | **Stack-RNN** | **Tape-RNN** | **Transformer** | **LSTM** | **Deltaformer** |
| **R** | Modular Arithmetic (Simple) | **100.0** | **100.0** | **100.0** | 24.2 | **100.0** | **100.0** |
| | Parity Check | **100.0** | **100.0** | **100.0** | 52.0 | **100.0** | **100.0** |
| **DCF** | Stack Manipulation | 56.0 | **100.0** | **100.0** | 57.5 | 59.1 | 58.3 |
| | Reverse String | 62.0 | **100.0** | **100.0** | 62.3 | 60.9 | 63.4 |
| **CS** | Duplicate String | 50.3 | 52.8 | **100.0** | 52.8 | 57.6 | 54.7 |
| | Odds First | 51.0 | 51.9 | **100.0** | 52.8 | 55.6 | 52.5 |

Table 5: Language hierarchy solvable by different model architectures

# E    Efficient Chunk-wise Implementation

Below is a simple PyTorch implementation, serving as pseudo-code. We can easily modify the selection of the kernel function or remove the normalization term. We tried three different ways of running time on the H100, as shown in Table 6. And we can see that the chunkwise algorithm has a 22x speed improvement compared to the recurrent implementation. At the same time, compared to fully parallel algorithms, it has an 8x speed improvement, because fully parallel algorithms are bounded by I/O, due to the $n \times n$ size matrix. The details can refer to the Readme file in the supplementary materials of the Triton implementation.

| Method | Time |
|---|---|
| Recurrent | 279.9 ms |
| Parallel | 102.2 ms |
| Chunk-wise | 12.7 ms |

Table 6: Comparison of execution times with tensor shape [2,32,8192,128] in an H100.

```python
import torch
import torch.nn.functional as F
import math

def flash_attn(K_chunk, K_prev, V_prev):
    attn = K_chunk @ K_prev.transpose(-1, -2)/math.sqrt(K_chunk.shape
    [-1])
    z_intra = torch.logsumexp(attn, dim=-1)
    return torch.softmax(attn,dim=-1)@V_prev, z_intra

def naive_implementation(k, n, d_model): """n is the previous v, v is
    actually new v. """
    B, H, T, D = k.shape
    v = torch.zeros_like(n)
    for t in range(T):
        if t == 0:
            v[:, :, 0] = n[:, :, 0]
        else:
            scores = torch.matmul(k[:, :, :t], k[:, :, t].unsqueeze
    (-1)).squeeze(-1) / math.sqrt(d_model)
            attn_probs = F.softmax(scores, dim=-1)
            v[:, :, t] = n[:, :, t] - torch.sum(attn_probs.unsqueeze
    (-1) * v[:, :, :t], dim=-2)
    return v

def optimized_chunked_implementation(K, N, d_model, C):
    B, H, T, D = K.shape
    V = torch.zeros(B, H, T, D)
    chunk_nums = T // C
    mask = torch.tril(torch.ones(C, C),diagonal=-1).unsqueeze(0).
    unsqueeze(0).to(K.device)
    for chunk_num in range(chunk_nums):
        start = chunk_num * C
        end = (chunk_num + 1) * C
        K_chunk = K[:, :, start:end]
        N_chunk = N[:, :, start:end]
        if chunk_num > 0:
            intra_output, Z_intra = flash_attn(K_chunk, K[:, :, :start
    ], V[:, :, :start])#O(TCD)
            A = (K_chunk @ K_chunk.transpose(-2, -1)).masked_fill(mask
    [:, :, :C, :C] == 0, float("-inf"))  / math.sqrt(d_model)#O(C^2D)
            Z_inter = torch.logsumexp(A, dim=-1)
            P = N_chunk - intra_output * (1/(1 + torch.exp((Z_inter-
    Z_intra).unsqueeze(-1))))
            A = F.softmax(A, dim=-1) * (1/(1 + torch.exp((Z_intra-
    Z_inter).unsqueeze(-1))))
            A[:,:,0,:] = 0
        else:
            A = (K_chunk @ K_chunk.transpose(-2, -1)).masked_fill(mask
    [:, :, :C, :C] == 0, float("-inf"))  / math.sqrt(d_model)
            A = F.softmax(A, dim=-1)
            A[:,:,0,:] = 0
            P = N_chunk
        Ti = torch.eye(C).unsqueeze(0).unsqueeze(0).unsqueeze(0).to(K.
    device) + A
```

```
      Ti_inverse = torch.inverse(Ti) ## Forward substitution method
   O(C^3) Each block can be solved in parallel if we don't use the
   normalization of softmax.
      V[:, :, start:end] = Ti_inverse @ P       # O(C^2D)
   return V     #O(T/C * (TCD + C^2D)) = O(T^2D + TCD + TC^2)

def verify_equivalence():
    B = 2
    H = 2
    T = 1024
    D = 64
    C = 32
    K = torch.randn(B, H, T, D)
    N = torch.randn(B, H, T, D)
    naive_output = naive_implementation(K, N, D)
    optimized_output = optimized_chunked_implementation(K, N, D, C)
    equivalence = torch.allclose(naive_output, optimized_output, atol
    =1e-5)
    print(f"{equivalence}")
```

Listing 1: PyTorch-style pseudo-code.

# F   The stability of the calculation of $u$ and $o$

We rewrite the calculations for $u$ and $o$ as follows:

$$u = A_1^{-1}v$$
$$o = A_2 u, \tag{31}$$

where $A_1(i,j) = \kappa_1(k_i, k_j)$, $A_2(i,j) = \kappa_2(k_i, k_j)$.

Then we will have:

$$\|(A_1 + \Delta A)^{-1}V - A_1^{-1}V\| \approx \|A_1^{-1}(\Delta A)A_1^{-1}V\| \leq \|A_1^{-1}\|\|\Delta A\|\|A_1^{-1}\|\|V\| = \|A_1^{-1}\|^2\|\Delta A\|\|V\|, \tag{32}$$

and

$$\|(A_2 + \Delta A_2)U - A_2 U\| = \|(\Delta A_2)U\| \leq \|\Delta A_2\|\|U\|. \tag{33}$$

The stability of the calculation for $u$ is weaker than that for $o$, so the selection of the $\kappa_1$ need to balance stability and expressivity.

# G   Code for synthetic data.

Here we provide a code for synthesizing data and the encoding of input information.

### G.1   Track the Exchange of Elements.

```
import numpy
import random
n_elements = 5
swap_pairs = [(i, j) for i in range(n_elements) for j in range(i+1,
    n_elements)]

def apply_swap(perm, swap_idx):
    i, j = swap_pairs[swap_idx]
    perm = list(perm)
    perm[i], perm[j] = perm[j], perm[i]
    return tuple(perm)

def generate_data(k, num_samples):
    data = []
```

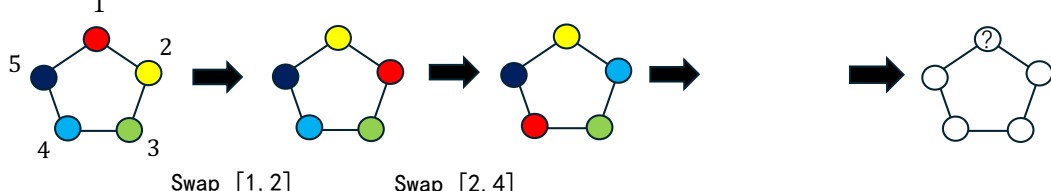

Figure 7: Swap task diagram. At the beginning, tokens of different colors are placed at positions 1 to 5, and the tokens of two positions are exchanged at each step. We expect the model to query what the token for each position is at each step. Simply but without loss of generality, we default to outputting the token at the first position to avoid introducing a "query token". This task can also be tokenized into a task with an input vocabulary size of $C_5^2 = 10$ and an output vocabulary size of $5$.

```
for _ in range(num_samples):
    swap_sequence = [random.randint(0, len(swap_pairs)-1) for _ in
 range(k)]
    current_perm = tuple(range(n_elements))
    first_elements = []

    for swap_idx in swap_sequence:
        current_perm = apply_swap(current_perm, swap_idx)
        first_elements.append(current_perm[0])

    input_ids = torch.tensor(swap_sequence, dtype=torch.long)
    labels = torch.tensor(first_elements, dtype=torch.long)
    data.append((input_ids, labels))

return data
```

## G.2 Reachability of directed acyclic graphs.

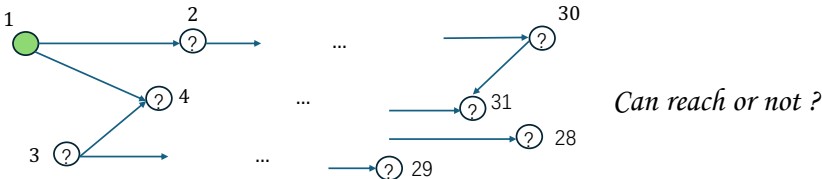

Figure 8: Reachability of directed acyclic graphs. Each node encodes at most its neighboring node information at the beginning. Then the model need to determine whether a node can reach from a starting point.

```
import numpy
import random
import torch.nn as nn
def create_graph(n):
    if n % 2 != 0:
        raise ValueError("n should be an even number")

    # Step 1: Randomly divide the points into two sets S_1 and S_2
    points = list(range(1, n + 1))
    random.shuffle(points)
    mid = n // 2
    S_1, S_2 = sorted(points[:mid]), sorted(points[mid:])

    def assign_parents(S):
        parents = {}
```

```python
        for i in range(1, len(S)):
            possible_parents = S[:i]
            parents[S[i]] = random.choice(possible_parents)
        return parents

    # Step 2: Assign parent nodes within each set
    parents_S1 = assign_parents(S_1)
    parents_S2 = assign_parents(S_2)

    # Step 3: Build adjacency matrix
    adjacency_matrix = np.eye(n)
    def fill_adjacency_matrix(parents):
        for child, parent in parents.items():
            if parent is not None:
                adjacency_matrix[child - 1][parent - 1] = 1

    fill_adjacency_matrix(parents_S1)
    fill_adjacency_matrix(parents_S2)
    labels = [0 for i in range(n)]
    if 1 in S_1:
        for i in S_1:
            labels[i-1] = 1
    else:
        for i in S_2:
            labels[i-1] = 1
    return labels, adjacency_matrix

def generate_graph_data(num_samples=100, n=32):
    """
    Generates graph data samples with reachability information.

    :param num_samples: Number of samples to generate.
    :param n: Number of nodes in the graph.
    :return: A list of tuples. Each tuple contains an adjacency matrix
     and a list of labels indicating reachability from node 1 to each
    node.
    """
    data = []
    for _ in range(num_samples):
        labels, A = create_graph(n)
        adj_matrix = torch.tensor(A, dtype=torch.float)
        # adj_matrix  =  adj_matrix.transpose(0,1)
        labels = torch.tensor(labels, dtype=torch.long)
        data.append((adj_matrix, labels))
    return data
class Emb(nn.Module): #Encode the neighbor node information of each
    node and mark the starting point
    def __init__(self, config):
        super().__init__()
        self.hidden_size = config.hidden_size
    def forward(self, x):
        # x shape: (batch_size, seq_len, input_dim)
        batch_size, seq_len, input_dim = x.shape
        pos_onehot = torch.zeros(seq_len, seq_len, device=x.device)
        pos_onehot[0, 0] = 1  # Mark the starting point
        pos_emb = pos_onehot.unsqueeze(0).expand(batch_size, -1, -1)
    # (batch_size, seq_len, seq_len)
        current_dim = x.size(-1)
        if current_dim < self.hidden_size:
            pad_size = list(x.shape)
            pad_size[-1] = self.hidden_size - current_dim
            padding = torch.zeros(*pad_size, device=x.device)
            x = torch.cat([x, padding], dim=-1)  # (batch_size,
    seq_len, hidden_size)
        return x.to(dtype)
```

