# OpenReview forum: "DeltaFormer: Unlock the state space of Transformer"
_NeurIPS.cc/2025/Conference — NeurIPS 2025 poster_

### Official Review · Reviewer_PXWq · 2025-06-30

**Clarity:** 1
**Significance:** 1
**Originality:** 2
**Rating:** 5
**Confidence:** 5

**Summary:**

This work proposes "DeltaFormer", a generalization of previously proposed "DeltaNet" sequence models. Concretely, the authors interpret the "DeltaNet" state update and readout mechanisms in terms of kernel functions, which allows them to implicitly operate in infinite-dimensional Hilbert spaces. The "DeltaNet" equations (5) and (6) are generalised by use of two different kernel functions and two different gates in (7) and (8). The authors show that their method can emulate an ${NC}^1$ complete problem both in theory and experimentally, they propose a chunk-wise parallel algorithm for their method, and they finally test the proposed method on realistic language modelling tasks.

**Questions:**

I do not have any immediate questions. All addressable points are listed as weaknesses. I hope you find this review constructive.

**Ethical Concerns:**

["NO or VERY MINOR ethics concerns only"]

**Final Justification:**

The authors have shown significant engagement during the discussion period. They have clarified and expanded on a range of important aspects of their work, and I believe that they successfully demonstrated the benefits of their approach. With the large amount of additional work done to the paper, I believe this work now merits acceptance.

**Limitations:**

"yes"

**Quality:**

2

**Strengths And Weaknesses:**

Strengths:

- The method is clearly superior to the transformer on the element exchange state tracking task and DAG reachability. Considering that you found it manageable to conduct medium-scale NLP experiments, it seems that it could maybe be used in larger models too.

- The chunk-wise matrix inversion method could offer a more optimal trade-off between recurrence and parallelism compared to the naive full recurrence. This is clearly applicable to any such linear "time-mixing" operator.

- It is an interesting observation that kernel choice influences the performance on synthetic tasks so much more than on more "natural" ones.

Weaknesses:

- As a general note, the paper could be written with better style and clarity. For example, lines 32-33 state that "it has been proven that in the ${TC}^0$ class, chain of thought needs to be utilised to solve problems in larger classes". I understand that CoT allows log-precision transformers to emulate circuits whose complexity is beyond that of ${TC}^0$. The quoted sentence seems to try to express this, but it is not clearly formulated (**within** ${TC}^0$, CoT is needed **to escape** ${TC}^0$?) . I don't want to penalise non-native speakers for imperfect English; what I am talking about here are incorrectly expressed ideas.

- The paper really could benefit from a report on the speed-up of the chunk-wise matrix inversion method as compared to the recurrent implementation.

- It was not made clear why "swap" is ${NC}^1$ complete under ${AC}^0$ reductions. At least you should point to literature showing that the permutation group over 5 elements is non-solvable, hence ${NC}^1$-complete.

- Considering that you generalised the readout function, it would have made sense to ablate the $\kappa_2$ design on retrieval tasks too.

- It seems that it makes the most sense to simply stick with the linear kernel, in which case we revert back to the standard delta-net.

- The theorems and proofs could be better explained. It took me a disproportionate amount of time to parse.

- There was no comparison to any related work excluding the Transformer in the experimental section.

---

> ### Author Response · Authors · 2025-08-01
> **Rebuttal (1/2)**
>
> Thank you very much for your constructive suggestions on our work. We hope the following response can solve your concerns.
>
> 1.**The paper could be written with better style and clarity.**
>
> Thank you for pointing this out. We sincerely apologize for the unclear expressions that led to confusion. We have carefully revised the corresponding parts in the revised manuscript to better convey our intended ideas. We greatly appreciate your feedback, which helped us improve the clarity of our presentation.
>
> 2.**The paper really could benefit from a report on the speed-up of the chunk-wise matrix inversion method as compared to the recurrent implementation.**
>
> Thank you very much for your valuable suggestion.  We compare it in our supplementary materials currently, and we will compare it in subsequent versions within the main text.
>
> | Method  (tensor shape = [2,32,8192,128])             | Time  |
> |----------------------|-------|
> | Recurrent            | 279.9 ms |
> | Parallel             | 102.2 ms |
> | Chunk-wise           | 12.7 ms |
>
> We can see that the chunkwise algorithm has a 22x speed improvement compared to the recurrent implementation. At the same time, compared to fully parallel algorithms, it has an 8x speed improvement, because fully parallel algorithms are bounded by I/O, due to the n * n size matrix. The details can refer to  the Readme file in the supplementary materials of the Triton implementation.
>
> 3.**It was not made clear why "swap" is NC1 complete under AC0 reductions.**
>
> Thank you for your suggestion. Firstly, starting from literature [1], the permutation group over 5 elements is non-solvable, hence NC1-complete.  Then permutations and several swaps can be converted to each other easily, for example, (a1 a2 a3 a4) (a5)=(a1 a2) (a1 a3) (a1 a4) (a5). Therefore, only considering the swap, it is still NC^1- complete.
> [1] Bounded-width polynomial-size branching programs recognize exactly those languages in nc. pp. 1–5, 1986.
>
> 4.**Considering that you generalised the readout function, it would have made sense to ablate the $\kappa_2$  design on retrieval tasks too.**
>
> Thank you very much for your suggestion. We conducted corresponding experiments on MQAR[1]. The setting is            vocab_size=256, input_seq_len=128,num_kv_pairs=32,d_model=32. And we controlled $\kappa_1$ to be linear to avoid that it enhence the ability of retrieve. The experimental results are as follows. We can find that the retrieval ability of linear is very poor.
>
> | $\kappa_2$  | Accuracy |
> | ----------- | -------- |
> | **Linear**  | 85.6     |
> | **Round**   | 91.6     |
> | **ReLU**    | 99.5     |
> | **Softmax** | 99.1     |
>
> [1] Zoology: Measuring and Improving Recall in Efficient Language Models
>
> 5.**It seems that it makes the most sense to simply stick with the linear kernel, in which case we revert back to the standard delta-net.**
>
> We  compared linear and nonlinear in Appendix B.1. It can be found that when the number of elements to be tracked exceeds the head dim, the linear method will have severe degradation.  We must acknowledge that if our future tasks are not so difficult and the model only needs to track the movement of chess pieces on the board, then a linear kernel with head dim=128 is sufficient. But if we want to track the movement of tens of thousands of objects. Instead of using a linear kernel, it's better to try a non linear deltaformer.
>
> 6.**The theorems and proofs could be better explained. It took me a disproportionate amount of time to parse.**
>
> Lemma1 of rwkv7 proves that the linear form of delta rule can achieve the ability to track element swapping, and it uses head dim $O (n)$ to complete its proof.  Moreover, due to the finite state space that is displayed and easy to understand, the entire proof process is relatively simple.
>
> And our Theorem 1 proves that even when we use the no-linear delta rule, we can still achieve the ability to track element swapping, and the required head dim changes from $O (n)$ to $O (log n)$. Due to nonlinearity, the state space cannot be written explicitly, making the entire proof process complex.
>
> The key to proving the Theorem 1 is that there are quite a few almost orthogonal directions in n-dimensional space, and when we use nonlinear functions as a new similarity measure, these directions become orthogonal under the new measure.  And the proof process adopts mathematical induction, mainly to prove that at each step, implicitly, the positions participating in the exchange are exchanged, while the positions not participating in the exchange remain unchanged.

---

> ### Author Response · Authors · 2025-08-01
> **Rebuttal (2/2)**
>
> 7. There was no comparison to any related work excluding the Transformer in the experimental section.
>
> a) **Yes, that was our initial consideration.**
>    We believed that for highly non-linear RNNs, their performance in state-tracking related tasks would be satisfactory. A significant advantage for Deltaformer would become apparent through comparisons in retrieval tasks. However, since retrieval tasks are not the main focus of this paper, we did not extend our comparisons beyond Transformer-based models.
>
> b) We are, however, very grateful for your insightful feedback. It highlighted the considerable value of conducting a more extensive comparison, which led us to perform experiments on synthetic datasets. The experimental outcomes are presented below:
>
> | Model                         | Parity |
> |------------------------------|--------|
> | Transformer                  | 0.022  |
> | Mamba [0,1]                  | 0.000  |
> | Mamba [-1,1]                 | 1.000  |
> | DeltaNet [0,1]               | 0.017  |
> | DeltaNet [-1,1]              | 1.000  |
> | mlstm                        | 0.087  |
> | slstm                        | 1.000  |
> | **Deltaformer Linear + softmax** | **1.000** |
> | Deltaformer softmax + softmax | 0.021 |
> | **Deltaformer exp + softmax** | **1.000** |
> | **Deltaformer ReLU + softmax** | **1.000** |
> | Deltaformer round + softmax, β∈[0,2] | 0.037 |
> | **Deltaformer round + softmax, β∈[-1,2]** | **1.000** |
> | Deltaformer round + round    | 0.041  |
>
> c) The reason we couldn't compare on language modeling tasks is that models around 400M are too small and exhibit significant bias, leading to unreliable experimental results. Larger-scale experiments would require more computational resources and infrastructure adaptation. Therefore, we only trained a **14B total-activation MoE model for 500B tokens** to compare Transformer and Deltaformer. The experimental results are as follows, and we hope they provide some insight.
>
> #### Benchmark (accuracy)
>
> | Dataset       | Transformer | DeltaFormer |
> |---------------|-------------|-------------|
> | COPA          | 73.8        | 74.6        |
> | ARC-E         | 80.9        | 82.9        |
> | ARC-C         | 50.8        | 51.2        |
> | PIQA          | 78.2        | 79.4        |
> | C-Eval        | 44.0        | 46.3        |
> | MMLU          | 43.8        | 45.2        |
> | RACE-High     | 48.8        | 49.0        |
> | RACE-Middle   | 62.5        | 62.9        |
> | SIQA          | 55.6        | 54.7        |
> | Winogrande    | 64.6        | 67.2        |
> | **Average**   | **60.3**    | **61.34**   |
>
> #### Training Tokens & Loss
>
> | Tokens | General Domain Loss | Code Domain Loss |
> |--------|---------------------|------------------|
> |        | Transformer | DeltaFormer | Transformer | DeltaFormer |
> | 100 B  | 2.06256     | 2.06254     | 1.52628     | 1.48656     |
> | 200 B  | 1.94337     | 1.94336     | 1.42112     | 1.37757     |
> | 300 B  | 1.88013     | 1.87554     | 1.35612     | 1.32587     |
> | 400 B  | 1.83119     | 1.83558     | 1.32285     | 1.29567     |
> | 500 B  | 1.81472     | 1.81032     | 1.31231     | 1.28326     |
>
> a) **Firstly, the benchmark results show that Deltaformer outperforms the baseline.**
> b) **Then there is the result of training loss**, which leads by **0.003** on the general domain, basically aligning with the slight increase of **3%** in FLOPs. However, in the code domain, the loss leads by **0.05**. When training 300 B tokens, the code loss can match the baseline training at 400 B, which far exceeds the gain from FLOPs. We believe this is due to higher expressiveness.

---

> > ### Comment · Reviewer_PXWq · 2025-08-04
> >
> > I would like to thank the authors for their comprehensive feedback. It seems that several results which I originally believed were missing from the paper were in fact present in the appendix, these being the runtime measurements and the ablation in Appendix B.1.
> >
> > I have also considered the extensive responses by the authors to the other reviewers. I believe the additional results add significantly to the value of the paper, so I will raise my score to accept.

---

> > > ### Author Response · Authors · 2025-08-04
> > >
> > > Thanks very much for reviewing our response and raise the score！

---

### Official Review · Reviewer_scvB · 2025-06-30

**Clarity:** 2
**Significance:** 3
**Originality:** 2
**Rating:** 5
**Confidence:** 4

**Summary:**

This paper proposes a new architecture DeltaFormer that draws inspiration from both transformers and DeltaNet. Drawing from the expressivity literature, the authors claim their architecture should have greater expressivity for state tracking, a task shown to be likely beyond transformers. The authors also evaluate their new architecture on targeted synthetic tasks, finding boosts on state tracking and graph connectivity. They also show that their model achieves similar performance to transformers on standard LM evaluations.

**Questions:**

> Another noteworthy fact is that existing large models have demonstrated amazing reasoning abilities after reinforcement learning [25, 24, 59], recent studies have shown that models have not unlocked new abilities through RL, and their abilities are limited by pre-training [73, 60].

Note that these models being trained with RL are using CoT, which means they have expressive power outside TC0.

**Ethical Concerns:**

["NO or VERY MINOR ethics concerns only"]

**Final Justification:**

As mentioned in my comment below, the new empirical results, isoFLOPs discussion, and careful elaboration of the theoretical results have addressed many of my initial reservations, beyond the lack of a hybrid model baseline. That said, one paper cannot do everything, and I think this paper would still be a valuable data point for the community on the potential of more expressive language models. Ideally, the others should acknowledge the lack of comparison to hybrid models (a simpler way to gain expressivity beyond transformers) as a limitation.

**Limitations:**

As discussed in weaknesses, the paper would benefit from comparing against a hybrid model (simply mixing DeltaNet layers with transformer layers). The presentation of the greater expressivity of DeltaFormer was also unclear, so I am not confident that its ability to solve NC1-complete problems is correct.

**Paper Formatting Concerns:**

TC0 formatting is inconstitent between paper and abstract. Also nit: TC^0 -> \mathsf{TC}^0

"which is much smaller 56 then linear kernel." -> "than a"

**Quality:**

2

**Strengths And Weaknesses:**

## Strengths

1. This paper addresses the theoretically motivated problem of developing an architecture with expressivity beyond TC0, proposing a new, likely more expressive architecture by integrating DeltaNet with transformers.
2. The empirical evaluation of the architecture is extensive, spanning both synthetic evaluations on targeted tasks of theoretical interest and general language modeling evaluations.

## Weaknesses

### Missing Baselines and Comparisons

Two important questions that should be answered to justify a new architecture are its compute efficiency and whether its complexity is justified compared to a simpler approach.

Regarding compute efficiency, DeltaFormer achieves similar numbers to the transformer in Table 1. But how does it compare in compute efficiency, i.e., are the models matched in number of parameters or in FLOPs? If you're adding some more expensive operations like matrix inversion, then it could conceivably have the effect of making the architecture less compute-efficient.

Regarding potentially simpler approaches, ow would DeltaFormer compare, both theoretically and empirically, to the simpler approach of mixing DeltaNet layers with Transformer layers? It would be good to have this comparison, and also potentially consider whether you hypothesize or can theoretically show any expressivity differences between DeltaFormer and a hybrid model. In particular, I think this comparison to hybrid architectures would be important to say something about.

### Clarity and Potential Correctness Issues

Figure 2 is potentially misleading. What does it mean to put LSTMs and RNNs at the level of P? They can't represent all of P, so presumably you intend this to mean they can solve some P-complete problem. If so, you should say this and also add a reference for which P-complete problem can be solved by RNNs.

Theorem 1 is not clear. This part should be reorganized to explicitly state which NC1-complete problem is solvable and how this relates to the construction. The following claim should also be made more precise:

> In summary, as a general form of transformer, DeltaFormer not only does it surpass the inherent TC0 expressivity of the Transformer, but it can also track the exchange of n objects using much smaller state space than models such as RWKV7.

Another concern is with the experimental setup in the state tracking experiments:

> And the 1-layer 194 DeltaFormer can execute and track the exchange operations of 5 elements. But increasing the number 195 of layers in the transformer did not improve either.

This is different from the findings here: https://arxiv.org/abs/2503.03961, where Theta(log n) depth allowed transformers to solve A5 up to length n. This suggests that there might truly be optimization issues involved in the failure of the transformers.

Relatedly:

> We speculate that this is likely the fundamental reason why existing large-scale models achieve randomness in entity tracking tasks [12].

Though with log n depth, the mentioned paper shows saturated-attention transformers can express state tracking up to contexts of length n (even if not for very long contexts).

> Due to the fact that matrix inversion is a task within NC2

How do you know this? Add a proof or reference

---

> ### Author Response · Authors · 2025-08-01
> **Rebuttal (1/2)**
>
> Thank you very much for recognizing our work in addressing motivation and conducting evaluations from synthesis tasks to language modeling. At the same time, you have also raised some concerns, and we hope that our following response can solve your concerns.
>
> 1.Missing Baselines and Comparisons
>
> Regarding compute efficiency,  the number of parameters can be perfectly aligned, but the flops will slightly increase. On the 340m scale model, approximately 5% of flops were added. Later, we conducted experiments on the 14b activated MOE model, where the number of kv was 1/4 of the number of q. As a result, the flops in the self-attention section increased by 25%, and in the entire MOE model, the flops increased by 3%. Some interesting large-scale experimental results are listed here:
>
> | Benchmark   | Transformer | DeltaFormer | Δ         |
> | -| - | - | - |
> | COPA        | 73.8        | 74.6        | +0.8      |
> | ARC-E       | 80.9        | 82.9        | +2.0      |
> | ARC-C       | 50.8        | 51.2        | +0.4      |
> | PIQA        | 78.2        | 79.4 | +1.2      |
> | C-Eval     | 44.0        | 46.3  | +2.3      |
> | MMLU  | 43.8        | 45.2  | +1.4      |
> | RACE-High   | 48.8        | 49.0  | +0.2      |
> | RACE-Middle | 62.5  | 62.9 | +0.4      |
> | SIQA        | 55.6 | 54.7  | −0.9      |
> | Winogrande  | 64.6  | 67.2 | +2.6      |
> | **Average** | **60.3**    | **61.34**   | **+1.04** |
>
> | Training Tokens | General Domain Loss | |Code Domain Loss |  |
> | - | - | - | -| -|
> |  | **Transformer**     | **DeltaFormer**  | **Transformer** | **DeltaFormer** |
> | 100 B | 2.06256 | 2.06254  | 1.52628 | 1.48656  |
> | 200 B  | 1.94337 | 1.94336 | 1.42112 | 1.37757 |
> | 300 B | 1.88013 | 1.87554 | 1.35612  | 1.32587  |
> | 400 B | 1.83119  | 1.83558| 1.32285  | 1.29567  |
> | 500 B | 1.81472  | 1.81032 | 1.31231 | 1.28326 |
>
>
>
> a) Firstly, the benchmark results show that the deltaformer outperforms the baseline
> b) Then there is the result of training loss, which leads by 0.003 on the general domain, basically aligning with the slight increase of 3% in flops. However, on the code domain, the loss leads by 0.05. When training 300b. The training token can match the baseline training of 400b, which far exceeds the gain of flops. We believe this is due to higher expressiveness.
> By the way, we are still working hard on optimizing training efficiency. There is an implementation of chunk wise that we wrote using Triton in the attachment. Its speed is currently only 1/3 of flash attention at 8k training length. But it is 22 times faster than naive recurrent implementation. As for the decode stage, the computation logic of deltaformer and transformer is similar, and in the case of I/O bound, there is almost no increase in time overhead.
>
> Comparisons with hybrid models:  the simpler approach of mixing DeltaNet layers with Transformer layers.
>
> We must acknowledge that this is a very good question, with both theoretical and practical significance!
> 1. Firstly, theoretically speaking, this simple mixing, or mixing using positional encoding like path, can only be proven to track the exchange of n objects using O (n) head dim, which is constrained by the linear delta rule And we can rigorously prove that using a nonlinear kernel function, we can use the head dim of O (log n) to track the exchange of n objects
> 2. In practice, a) we can prove the fact that when the number of objects to be tracked, n, exceeds the head dim, the performance of a simple hybrid architecture will significantly degrade in the synthetic data of the swap task, which is similar with the result in Appendix B.1  b) In small-scale language modeling tasks with 340m parameters, no significant differences can be seen. c) In larger scale tasks, more resources are needed for comparative experiments. We will verify it when we have sufficient computing resources in the future.
> 3. In the end, we argue that a formulation grounded in more principled foundations is more elegant than a simple hybrid.
>
> And their purpose of mixing is to improve the insufficient retrieval ability of linear models. From this perspective, simple mixing models are successful. If it is to track the exchange of exponential objects, simple mixing is not effective. We must also acknowledge that if our future tasks are not so difficult and the model only needs to track the movement of chess pieces on the board, then a delta with head dim=128 is sufficient. But if we want to track the movement of tens of thousands of objects, so instead of using a linear deltaet, it's better to try a non linear deltaformer.
>
> 2.Figure 2 is potentially misleading.
>
> You're absolutely right that being in a certain class doesn't necessarily mean it can solve all the problems of that class. This diagram wants to convey is that within a certain class, it means that it is not restricted by classes lower than this class, and it can solve some problems within this class. We will clarify this in the future to avoid misunderstanding.

---

> ### Author Response · Authors · 2025-08-01
> **Rebuttal (2/2)**
>
> 3.**Clarification of Theorem 1 and Related Statements **
>
> Theorem 1 (rephrased)
> Let the list be $L=[1,2,\dots,n]$.
> Define the swap operation $G_{i,j}$ as
>
> $$G_{i,j}(L)[k]=
> \\begin{cases}
> L[j], k=i, \\\\
> L[i], k=j, \\\\
> L[k], \text{otherwise}
> \\end{cases}$$
>
> Let a sequence of swaps be $g_1,g_2,\dots,g_l \in G = \\{ G_{i,j}| 0<i<j \leq n \\} $.
> Then there exists a single-layer DeltaFormer $F$ with head dimension $O(\log n)$ such that
>
> $$F(g_1,\dots,g_l,\, j)=g_1\circ g_2\circ\cdots\circ g_l(L)[j].$$
>
> KU Cache read out and rewrite
>
> We can introduce the KU cache compress operator $\mathrm{ku}(\cdot)$ that pre-fills the sequence $g_1,\dots,g_l$ into a cache of size $O(n\log n)$, independent of $l$.
> With this cache we obtain another single-layer DeltaFormer $H$ satisfying
> $$H\bigl(\mathrm{ku}(g_1,\dots,g_l),j\bigr)=F(g_1,\dots,g_l, j),$$
>
> Comparison with Lemma 2 in RWKV-7
> Lemma 2 in RWKV-7 (rephrased)
> There exists a single-layer RWKV-7 block $F$
>
> - head dimension $O(n)$,
> - state-space size $O(n^2)$,
>
> such that
>
> $F(g_1,\dots,g_l,j)=g_1\circ g_2\circ\cdots\circ g_l(L)[j].$
>
> 4.**Concern  with the experimental setup in the state tracking experiments. Some work has proven that transformers with $O(\log n )$ depth can complete state tracking.This is different from the findings here: https://arxiv.org/abs/2503.03961, where Theta(log n) depth allowed transformers to solve A5 up to length n.
>
> a) We agree with your viewpoint that for limited contexts, transformers can achieve good state tracking by increasing the number of layers. However, in terms of circuit complexity, what we hope to achieve is that this model can be completed regardless of the size /context length of the problem.
>
> b) We carefully check the experimental differences, and find that https://arxiv.org/abs/2503.03961 use the curriculum learning mentioned in their Appendix F.  This means that the transformer has indeed encountered optimization difficulties, requiring us to carefully design the curriculum learning to unlock its state tracking capabilities within training length.  For example, in [1], 16 layers of transformers were used to perform state tracking with a context length of **32**, which is also much lower than $$O (log n)$$ and once the length exceed the training length, it fails in generalization.
>
> So, why don't we design a transformer that can naturally complete the state tracking tasks?
>
> [1] Implicit Language Models are RNNs: Balancing Parallelization and Expressivity.
>
> 5."Due to the fact that matrix inversion is a task within NC2" How do you know this? Add a proof or reference
>
> "Let $I(n),E(n),D(n),P(n)$, donate the parallel arithmetic complexity of inverting order n matrics, solving a system of n linear equations in n unknowns, computing order n determinants and finding the characteristic polynomials of order n matrices respectively. .... Indeed, we have $\log(n) \leq I(n),E(n),D(n),P(n)\leq \log^2(n)$" -- Fast parallel matrix inversion algorithms. 1985.
>
> And we know that Transformer with constant precision is within $AC^0$. And the difference between the deltaformer is the caculation of $u$, which can be treated as a matrix inversion, which can be simulated by a $NC^2$ circuit. So we can roughly say that Deltaformer with constant precision is within $NC^2$.We must admit that using $NC^2$ as the upper bound of Deltaformer may not be a very tight bound, so we present it in discussion section to remind readers and we hope this can inspire someone in the future.
>
> 6.**Note that these models being trained with RL use CoT, which means they have expressive power outside **$TC^0$**.
>
> 1. You are right. We are not challenge this. What we want to express here is that some recent reflection on RL suggests that RL has not been able to unlock new abilities and that the model’s capabilities are limited by pre-training. During pre-training, no $O(n)$-length pathway was trained, so in this context improving model performance without relying on CoT in pretraining would be meaningful. We think that this can serve as an small part of the induction to introduce our work.
>
> 2. We think this is a very good question regarding the role of CoT and model architecture.
>
> a) First, expanding the expressiveness of model with COT and enhancing the expressiveness of the model itself are not contradictory, but orthogonal.
>
> b) CoT also has its drawbacks - bandwidth bottleneck. We must map to the language space, which introduces a log V-bit bottleneck. And we can treat Deltaformer as an implicit thought.
>
> c) A stronger base model may achieve higher decoding efficiency even when using CoT. For example, to determine the connectivity of a DAG with n nodes and diameter D, according to https://arxiv.org/abs/2505.12514, a standard CoT may require decoding $O(n^2)$ token and a continuous-space CoT only requires $O(D)$ tokens, where $O(D)$ maybe  $O(n)$, and Deltaformer might require zero decoding tokens to answer the question correctly after prefill.

---

> > ### Comment · Reviewer_scvB · 2025-08-03
> >
> > I thank the reviewers for their thorough response. I appreciate the additional experiments and isoFLOPs discussion. My concerns about clarity and correctness of the theory have been largely addressed, assuming the elaborations can be incorporated in revisions. A comparison to a hybrid baseline is still lacking, which should be acknowledged as a limitation.
> >
> > Conditional on the new empirical findings and theoretical elaborations being carefully incorporated into the manuscript (and appropriate discussion of the missing hybrid baseline), I believe it would be valuable for the community for the paper to appear at NeurIPS. I will therefore raise my score to Accept.
> >
> > ## IsoFLOPs Comparison
> >
> > > Then there is the result of training loss, which leads by 0.003 on the general domain, basically aligning with the slight increase of 3% in flops. However, on the code domain, the loss leads by 0.05. When training 300b. The training token can match the baseline training of 400b, which far exceeds the gain of flops. We believe this is due to higher expressiveness.
> >
> > I appreciate the new experiments more thoroughly comparing transformers and DeltaFormers. These provide more empirical grounding for assessing to what degree performance differences come from more FLOPs vs. architectural differences. Visualizing performance as a function of FLOPs (e.g., Figure 3 in RWKV-7) could be an effective way to present this data.
> >
> > ## Hybrid Baseline
> >
> > The paper's argument would still be strengthened by adding **other baselines** beyond transformers such as a hybrid transformer + SSM approach, which is similar in spirit to DeltaFormer and likely easier from an implementation perspective.
> >
> > ## Theory
> >
> > Thanks for the elaboration on Theorem 1 and the comparison to RWKV-7. These details definitely need to be better conveyed in the main body of the paper. Please also make the discussed clarifications to Figure 2.
> >
> > > We must admit that using NC2 as the upper bound of Deltaformer may not be a very tight bound
> >
> > Tangential, but could it be possible to show an upper bound of TC1?
> >
> > ## Log Depth State Tracking
> >
> > You make a good point about the lack of curriculum learning in your setup vs the Merrill & Sabharwal paper. Thanks for the clarification, and it would be good to mention this when discussing your results.

---

> > > ### Author Response · Authors · 2025-08-04
> > >
> > > Thank you very much for increasing score!  We will include the comparison of hybrid architectures in the limitations and incorporate the facts clarified in the rebuttal into the revised version.
> > >
> > >
> > > > Tangential, but could it be possible to show an upper bound of TC1?
> > >
> > > This is a good question. We believe that the deltaformer is also within $TC^1$. The inversion of a lower triangular matrix can be easily performed using the divide-and-conquer formula for block matrix inversion, where the depth is $O(\log n)$. And each small matrix multiplication can be simulated in $O(1)$ time using TC circuits, giving a total depth of $O(\log n)$. Therefore, the deltaformer is within $TC^1$.

---

### Official Review · Reviewer_WV7U · 2025-07-03

**Clarity:** 3
**Significance:** 4
**Originality:** 3
**Rating:** 5
**Confidence:** 3

**Summary:**

To address the precision-parallelism trade-off, this paper revisits the delta rule through the lens of kernel functions and propose DeltaFormer, a model that implicitly assigns the state space to Transformer. By introducing kernelization, DeltaFormer offers a novel interpretation of delta updating rule, e.g., DeltaNet as a special case that uses a specific similarity function, and further generalizes the operations using separate kernel functions and gates. DeltaFormer demonstrated improved performance over the Transformer in some toy tasks, but only achieved comparable results in small-scale language modeling in this paper.

**Questions:**

1. Where do state space models, currently gaining attention as recurrent alternatives, fall in Figure 2? According to ‘The Illusion of State in State-Space Models’ paper, are they considered to be in the same category as Transformers?
2. In the Swap task, is it possible to use the same input and output vocabulary? The phrase “to avoid introducing a ‘query token’” is unclear—could you clarify what this means?
3. Theorem 1 omits positional embeddings, but later sections favor RoPE or NoPE. Does this imply those methods yield better results? The explanation in Lines 215–220 could be clearer. Also, were the results in Figure 4 obtained without any positional encoding?
4. Are all models in Figure 6b single-layer? If so, it would help to state this explicitly in the caption (e.g., “DeltaFormer (Single Layer)”).
5. Line 240: “Therefore, the operation of matrix inversion greatly improves the expressivity of the model.” — This claim needs more elaboration. Why/how does matrix inversion contribute to expressivity here?
6. While the proposed model underperforms in small-scale settings, the authors suggest potential in industrial-scale applications. However, given that the motivation hinges on increasing model expressivity, shouldn’t some advantage already appear in small-scale experiments?

**Ethical Concerns:**

["NO or VERY MINOR ethics concerns only"]

**Final Justification:**

The rebuttal has clarified the paper’s contributions, and I believe its value is now more evident. I maintain my initial score of Accept.

**Limitations:**

yes

**Paper Formatting Concerns:**

No concern

**Quality:**

3

**Strengths And Weaknesses:**

- Strengths
    1. The paper identifies a key limitation in current sequence modeling paradigms and presents a generalized framework that aims to address the issue with a well-motivated approach.
    2. The model shows promising feasibility in controlled toy tasks, supporting its potential.
    3. Implementation details, including the design choices for similarity functions and the use of curriculum learning, are well explained and reproducible.
- Weaknesses
    1. If DeltaFormer successfully resolves the trade-off it targets, the benefit should also manifest in small-scale language modeling; however, the experiments show no clear improvement over the Transformer baseline.
    2. The presentation is weak. There are multiple typos (Lines 26, 56, 202), grammatical errors (Lines 188, 215, 236–239), and missing references (Lines 210, 211). Some expressions may even convey the opposite of the intended meaning (e.g., Line 219).

---

> ### Author Rebuttal · Authors · 2025-07-31
>
> We sincerely thank you for the positive assessment of our work and for the detailed, constructive feedback. Below, we try to address each concern in turn and hope our clarifications can solve your concern.
>
> 1.**The synthesis task has benefits, but small-scale models do not have significant gains.**
> We appreciate this important observation. We conjecture that the limited improvements seen in synthetic tasks at small scale stem from two factors:
> a) the bulk of the pre-training distribution is dominated by shallow, memorization-heavy patterns, and
> b) the model capacity is insufficient for acquiring the deeper computational primitives that our architecture is designed to exploit.
>
> Therefore, we conducted experiments on a MOE model with 680m activation parameters and 14b total parameters, and trained a total of 500b tokens. The benchmark results are shown below:
>
> | Benchmark (accuracy)    | Transformer | DeltaFormer |
> |--------------|-------------|-------------|
> | COPA         | 73.8        | 74.6        |
> | ARC-E        | 80.9        | 82.9        |
> | ARC-C        | 50.8        | 51.2        |
> | PIQA         | 78.2        | 79.4        |
> | C-Eval       | 44.0        | 46.3        |
> | MMLU         | 43.8        | 45.2        |
> | RACE-High    | 48.8        | 49.0        |
> | RACE-Middle  | 62.5        | 62.9        |
> | SIQA         | 55.6        | 54.7        |
> | Winogrande   | 64.6        | 67.2        |
> | **Average**  | **60.3**    | **61.34**   |
>
> | Training Tokens | General Domain (loss) |              | Code Domain (loss) |              |
> |-----------------|----------------|--------------|-------------|--------------|
> |                 | Transformer    | DeltaFormer  | Transformer | DeltaFormer  |
> | 100 B           | 2.06256        | 2.06254      | 1.52628     | 1.48656      |
> | 200 B           | 1.94337        | 1.94336      | 1.42112     | 1.37757      |
> | 300 B           | 1.88013        | 1.87554      | 1.35612     | 1.32587      |
> | 400 B           | 1.83119        | 1.83558      | 1.32285     | 1.29567      |
> | 500 B           | 1.81472        | 1.81032      | 1.31231     | 1.28326      |
>
> An instructive pattern emerges from the training curves. On the aggregate corpus, DeltaFormer attains a 0.003 lower loss, while incurring ≈3 % more FLOPs. However, on the code subset the gap widens to 0.05. Mean while  DeltaFormer reaches after 300 B tokens the loss the baseline achieves after 400 B tokens. This gain is much more than FLOP increase alone would explain. Under the “compression ≈ intelligence” hypothesis, we therefore expect even larger DeltaFormers to unlock further improvements on reasoning-heavy domains such as code.
>
> 2.**The presentation is weak. **
>
> We sincerely thank the reviewers for highlighting that the exposition can be improved. In the revision we will perform a full proof-read, restructure several sections for clarity, and add intuitive figures and pseudo-code to enhance overall readability.
>
> 3.**Where is the state space model?**
>
> Mamba2, retnet, rwkv6, hgrn2 is in the same category as Transformers, Deltanet, rwkv7, gated deltanet is beyond the  category of Transformers. In order to define these methods more clearly, we may also establish another axis regarding memory capacity to isolate transformer and deltaformer from these finite state space methods.
>
> 4.**Details about the swap tasks.**
>
> It is possible to use the same input and output vocabulary, for example, directly using a large vocabulary that includes both input and output vocabulary.  We apologize for not making it clear in the main text about 'to avoid introducing a query token'. The background is that we originally wanted to identify what elements are at each possible position, but in order to express what elements are at each possible position, we need to introduce an additional n tokens to represent what elements are at a certain position now, where n is the number of positions. If we only fixedly query what the current element is at the first position, then there is no need to introduce n additional tokens. This is just a simplification made to reduce the difficulty of learning, but it does not change the position of this task within the parallel complexity class.
>
> 5.**About position embeddings.**
> We did not use any position embedding in Theorem 1. Additionally, we did not use any position encoding in Figure 4. But before conducting actual language modeling, we are curious about the difference between the NoPE used in Theorem 1 and the RoPE used in actual language modeling. Therefore, we conducted the experiment in Figure 5.  The experimental results are also quite interesting, so we have included them in the main text.
> We can observe that when using RoPE, the model can have faster convergence, but the upper limit is low. If Nope is used, although the convergence is slow, with the help of course learning, the final upper limit is very high. We hope this experiment can spark further discussion on the role of rethinking position encoding.
>
> 6.**Figure 6b.**
>
> Yes, it's a single layer. We will improve our presentation to make our paper clear.
>
> 7.**Why/How does matrix inversion enhance expressivity?**
>
> Matrix inversion broadens the representational power of the model for two key reasons.
>
> a.**Complexity-theoretic relevance**
>    Certain parallel-complexity classes—specifically NL and SL —are characterized by graph-reachability tasks on directed and undirected graphs, respectively. These tasks lie strictly beyond the reach of constant-depth threshold circuits (TC⁰), the class to which standard Transformers are limited.
>
> b.**Algebraic reduction to matrix inversion**
>    Connectivity queries can be encoded as reachability in the adjacency matrix **A**. The series
>    $$
>    I + \beta A + (\beta A)^2 + \dots = (I - \beta A)^{-1} $$
>    converges (for sufficiently small β) and its (i, j) entry is positive iff node *j* is reachable from node *i*. Hence, computing$ (I - \beta A)^{-1} $ solves the connectivity problem exactly. Because matrix inversion is not in TC⁰, incorporating it equips the model with strictly greater expressive capacity.
>
> In short, matrix inversion enables the model to solve connectivity tasks that are provably out of reach for pure Transformer architectures, thereby substantiating the claim that it enhances expressivity.
>
> 8.**About small-scale experiments.**
>
> See rebuttal in "1.**The synthesis task has benefits, but small-scale models do not have significant gains.**"

---

> > ### Comment · Reviewer_WV7U · 2025-08-07
> >
> > I appreciate the authors for thoroughly addressing my concerns regarding the experimental setup and the advantages of the proposed method. The additional experiments further support the promise of the approach.

---

### Official Review · Reviewer_cP7d · 2025-07-03

**Clarity:** 3
**Significance:** 3
**Originality:** 3
**Rating:** 5
**Confidence:** 3

**Summary:**

This work introduces DeltaFormer, an extension of the delta rule using kernel functions in order to increase the expressive power of standard Transformers. By incorporating nonlinear functions, the authors show that DeltaFormer can learn state-tracking tasks such as swapping elements in a sequence, while maintaining competitive performance in language modeling tasks compared to standard Transformers.

**Questions:**

- Which similarity function did you use in the language modelling experiments?

-- References --

[1] Songlin Yang, Bailin Wang, Yu Zhang, Yikang Shen, and Yoon Kim. Parallelizing Linear Transformers with the Delta Rule over Sequence Length. Advances in Neural Information Processing Systems, 36, 2024

[2] R. Grazzi, J. Siems, A. Zela, J. Franke, F. Hutter, and M. Pontil. Unlocking State-Tracking in Linear RNNs Through Negative Eigenvalues. In The Thirteenth International Conference on Learning Representations (ICLR’25). ICLR, 2025.

[3] Gregoire Deletang, Anian Ruoss, Jordi Grau-Moya, Tim Genewein, Li Kevin Wenliang, Elliot Catt, Chris Cundy, Marcus Hutter, Shane Legg, Joel Veness, et al. Neural Networks and the Chomsky Hierarchy. In The Eleventh International Conference on Learning Representations, 2023.

**Ethical Concerns:**

["NO or VERY MINOR ethics concerns only"]

**Final Justification:**

Most of my concerns have been addressed, especially the comparison to other sub-quadratic models, scaling experiments and formal language tasks

**Limitations:**

Yes

**Paper Formatting Concerns:**

No formatting issues.

**Quality:**

3

**Strengths And Weaknesses:**

**Strengths:**
- The paper proposes a new model that increases the expressivity of Transformers by applying the delta rule on the kernel space.
- The new architecture is comparable on language modelling tasks with standard Transformer models, while achieving better performance on a state-tracking task, namely permuting elements (swaps).
- The authors provide an efficient implementation of DeltaFormer.

**Weaknesses:**
- The authors compare DeltaFormer only to Transformers, however there are several other subquadratic models that are more expressive and excel on state-tracking tasks. Even DeltaNet [1] with negative eigenvalues in the state-transition matrix is able to solve state-tracking tasks such as parity or permutation groups [2] and performs competitively on the same language modelling tasks used in this submission. I recommend the authors to add the results from these methods in their benchmark results.
- The authors demonstrate increased expressivity only on a single task. I strongly encourage to support the theory with more experiments, e.g. on formal language tasks as proposed in [3].
- I think evaluating larger models is necessary for wider adoption of DeltaFormer in the long run. The model shows increased expressivity by adding simple similarity functions between keys, however it is not clear how this will impact the performance at larger scale.

*Minor:*
- Line 210: reference to appendix broken.
- The figures are sometimes hard to read, especially due to the small font size.
- For better comparison, I suggest to put all the methods' learning curves on the same plot.

---

> ### Author Rebuttal · Authors · 2025-07-31
>
> We sincerely thank you for your insightful feedback. The references you provided were invaluable in deepening our discussion and strengthening the manuscript. In the following, we address each of your concerns in turn.
>
> 1.**Regarding other quadratic methods that perform well on state-tracking tasks.**
> We agree that certain quadratic alternatives can indeed achieve competitive results on these benchmarks.
>
> A. In the original submission we did not benchmark them exhaustively, because our primary goal was to propose an architecture that improves the *asymptotic* complexity of the Transformer rather than to outperform every quadratic variant on every task. We view these quadratic methods as strong baselines, yet they are ultimately constrained by the same retrieval limitations inherent in the standard attention mechanism.
>
> B. We did, however, include a comparison between linear and non-linear variants in Appendix B.1 (the linear variant can be interpreted as a form of delta rule). As shown there, once the number of elements to be tracked exceeds the head dimension, the linear variant degrades sharply. This observation echoes earlier studies comparing vanilla and linear Transformers: both can retrieve information in principle, but the former remains superior on long-context “needle-in-a-haystack” tasks.
>
> C. To further solidify our empirical results, we augmented the evaluation suite of Paper 2 with the Parity task. The new results are summarized as follow:
> | Model / Variant                              | Parity Accuracy |
> |----------------------------------------------|-----------------|
> | Transformer                                  | 0.022           |
> | Mamba [0, 1]                                 | 0.000           |
> | Mamba [-1, 1]                                | 1.000           |
> | DeltaNet [0, 1]                              | 0.017           |
> | DeltaNet [-1, 1]                             | 1.000           |
> | mLSTM                                        | 0.087           |
> | sLSTM                                        | 1.000           |
> | DeltaFormer – Linear + Softmax,  β ∈ [0, 2]             | 1.000           |
> | DeltaFormer – Softmax + Softmax,  β ∈ [0, 2]             | 0.021           |
> | DeltaFormer – Exp + Softmax,       β ∈ [0, 2]           | 1.000           |
> | DeltaFormer – ReLU + Softmax, β ∈ [0, 2]              | 1.000           |
> | DeltaFormer – Round + Softmax, β ∈ [0, 2]    | 0.037           |
> | DeltaFormer – Round + Softmax, β ∈ [-1, 2]   | 1.000           |
> | DeltaFormer – Round + Round, β ∈ [0, 2]                | 0.041           |
>
>  We observe that the choice of kernel function significantly influences DeltaFormer’s performance on Parity. Among the combinations we tested, “round + round” and “softmax + softmax” underperform relative to others. We suspect this stems from optimization difficulties—likely a need for finer-grained hyper-parameter tuning or a re-scaling of the gate range—which we leave to future work.
>
> 2.**Additional experiments on formal-language tasks to further support our theory.**
>
> In the original submission we concentrated on synthetic tasks that capture the strict expressive boundary of TC⁰, specifically accessibility problems on S₅ and directed acyclic graphs (DAGs). This focus allowed us to demonstrate that our architecture can provably exceed this boundary, and we therefore did not include other formal-language benchmarks.
> We are grateful for the reviewer’s suggestion and the accompanying references. While their official implementation is provided in JAX, our codebase is PyTorch-based; we are currently re-implementing the tasks in PyTorch to ensure full reproducibility and fair comparison. Preliminary runs are under way, and we will update the manuscript with the new results as soon as they are ready—no later than the camera-ready deadline.
>
> 3.**Evaluation at larger scale**
>
> We fully agree that demonstrating scalability is essential for the broader adoption of DeltaFormer. To this end, we trained a 14 B-parameter MoE model with 680 M active parameters on 500 B tokens. The downstream results are summarized as follow:
>
> | Benchmark (accuracy)    | Transformer | DeltaFormer |
> |--------------|-------------|-------------|
> | COPA         | 73.8        | 74.6        |
> | ARC-E        | 80.9        | 82.9        |
> | ARC-C        | 50.8        | 51.2        |
> | PIQA         | 78.2        | 79.4        |
> | C-Eval       | 44.0        | 46.3        |
> | MMLU         | 43.8        | 45.2        |
> | RACE-High    | 48.8        | 49.0        |
> | RACE-Middle  | 62.5        | 62.9        |
> | SIQA         | 55.6        | 54.7        |
> | Winogrande   | 64.6        | 67.2        |
> | **Average**  | **60.3**    | **61.34**   |
>
> | Training Tokens | General Domain (loss) |              | Code Domain (loss) |              |
> |-----------------|----------------|--------------|-------------|--------------|
> |                 | Transformer    | DeltaFormer  | Transformer | DeltaFormer  |
> | 100 B           | 2.06256        | 2.06254      | 1.52628     | 1.48656      |
> | 200 B           | 1.94337        | 1.94336      | 1.42112     | 1.37757      |
> | 300 B           | 1.88013        | 1.87554      | 1.35612     | 1.32587      |
> | 400 B           | 1.83119        | 1.83558      | 1.32285     | 1.29567      |
> | 500 B           | 1.81472        | 1.81032      | 1.31231     | 1.28326      |
>
> An instructive pattern emerges from the training curves. On the aggregate corpus, DeltaFormer attains a 0.003 lower loss, while incurring ≈3 % more FLOPs. However, on the code subset the gap widens to 0.05. Mean while  DeltaFormer reaches after 300 B tokens the loss the baseline achieves after 400 B tokens. This gain is much more than FLOP increase alone would explain. Under the “compression ≈ intelligence” hypothesis, we therefore expect even larger DeltaFormers to unlock further improvements on reasoning-heavy domains such as code.
>
> 4.**Clarifications requested by the reviewers**
>
> We thank the reviewers for their careful reading and for highlighting points that need further clarity. Below we address the questions.
> about the language-modeling similarity function. In our language-modeling experiments we adopted the conventional softmax (q kᵀ) attention score because we intentionally designed DeltaFormer such that the standard Transformer is a strict special case. Concretely, if we set β = 0 and α = 1 in Eq. 7 and let the similarity function in Eq. 8 coincide with the standard softmax attention, the update rule collapses to the familiar single-head softmax attention of the Transformer (see the derivation in Appendix C.2). We will make this correspondence more explicit in the revision.

---

> > ### Author Response · Authors · 2025-08-04
> > **Addition result about formal language tasks as proposed in [3].**
> >
> > In recent days, we have attempted to reproduce the content in [3] using Python code. We report the accuracy of the relevant models in [3] and the results of the Deltaformer ($\kappa_1( \cdot )$ is linear, $\kappa_2( \cdot )$ is softmax, no position embedding) as follows:
> >
> > | Level | Task                          | RNN   | Stack-RNN | Tape-RNN | Transformer | LSTM  | Deltaformer |
> > |-------|-------------------------------|-------|-----------|----------|-------------|-------| -------- |
> > | **R** | Modular Arithmetic (Simple)   | **100.0** | **100.0**     | **100.0**    | 24.2        | **100.0** | **100.0**  |
> > |       | Parity Check                 | **100.0** | **100.0**     | **100.0**    | 52.0        | **100.0** | **100.0** |
> > | **DCF** | Stack Manipulation           | 56.0  | **100.0**     | **100.0**    | 57.5        | 59.1  | 58.3 |
> > |       | Reverse String                | 62.0  | **100.0**     | **100.0**    | 62.3        | 60.9  | 63.4  |
> > | **CS** | Duplicate String             | 50.3  | 52.8      | **100.0**    | 52.8        | 57.6  | 54.7  |
> > |       | Odds First                    | 51.0  | 51.9      | **100.0**    | 52.8        | 55.6  |  52.5 |
> >
> >
> >
> > The results show that in the category of regular (R) language, Deltaformer has better length generalization ability compared to Transformer, indicating that Deltaformer can learn circuits that are easily length generalized compared to Transformer. But for some more complex languages, such as deterministic context-free (DCF) and context-sensitive (CS) languages, the performance of Deltaformer and Transformer is not as good as RNN models, especially the Tape-RNN proposed in [3]. However, considering that RNN models have not yet been efficiently implemented on GPUs, we believe that Deltaformer is practical.
> >
> > [3] Gregoire Deletang, Anian Ruoss, Jordi Grau-Moya, Tim Genewein, Li Kevin Wenliang, Elliot Catt, Chris Cundy, Marcus Hutter, Shane Legg, Joel Veness, et al. Neural Networks and the Chomsky Hierarchy. In The Eleventh International Conference on Learning Representations, 2023.

---

### Note · Authors · 2025-08-13

We sincerely appreciate the reviewers for their thoughtful and insightful comments, which have helped us improve our manuscript. We have addressed each concern in detail in our individual responses. Below, we highlight key points from the discussion and outline planned revisions to the manuscript.

**Positive Opinion after Rebuttal**

1. **Reviewer cP7d**: "I will increase my score to accept.";

2. **Reviewer WV7U**: "I appreciate the authors for thoroughly addressing my concerns regarding the experimental setup and the advantages of the proposed method."

3. **Reviewer scvB**: "I believe it would be valuable for the community for the paper to appear at NeurIPS. I will therefore raise my score to Accept."

4. **Reviewer PXWq**: "I believe the additional results add significantly to the value of the paper, so I will raise my score to accept."

**Planned Revisions to the Manuscript**

1. **Expanded experiments**: we will include additional synthetic experiments which evaluates state tracking, formal language understanding, and retrieval capabilities.  And we will include a comparative analysis between Deltaformer and Transformer on a 14B MoE model.

2. **Implementation details**: we will include the benchmark results comparing the Triton-optimized chunk-wise algorithm with the naive recurrent implementation which is provided in the supplement temporarily.

3. **Improved clarity and discussion**: we will include the enhanced discussion in rebuttal and improve readability of the manuscript.

4. **Limitations**: we will explicitly discuss the lack of large-scale validation and compare to a simple hybrid linear DeltaNet and softmax attention, which will be included in the limitations section.

We believe these revisions will strengthen the manuscript. We would like to once again express our sincere gratitude to all the reviewers for their time, valuable feedback, and constructive suggestions, which have significantly improved our work.

---

### Decision · Program_Chairs · 2025-09-17

**Decision:**

Accept (poster)

**Comment:**

This paper proposes a more expressive variant of Transformer that uses the delta rule-based update rule (i.e., DeltaNet), which increases the theoretical expressivity of Transformers beyond TC0. This involves making the DeltaNet-based RNN into an "infinite dimensional" RNN (which is equivalent to a Transformer with a different sequence-mixing mechanism) through the perspective of kernel functions. The model is coupled with a parallel chunkwise algorithm that enables efficient training. Experiments on synthetic and real-world tasks demonstrate the effectiveness of this new Transformer variant.

On the plus side, the work overcomes an important (theoretical) limitation of Transformers, and is backed up by solid empirical experiments as well as efficient training algorithms. The idea itself is also novel and well executed. On the negative side, there were some baselines that were lacking, and many reviewers noted that clarity/presentation could be greatly improved. Some of these issues were addressed during the rebuttal, and all reviewers were unanimous in recommending acceptance. I concur. (The nontrivial issues that remain with regard to presentation prevent me from recommending a higher rating.)